# The zinc metalloprotein MigC impacts cell wall biogenesis through interactions with an essential Mur ligase in *Acinetobacter baumannii*

Jeanette M. Critchlow[1], Joseph S. Rocchio[2], Melanie C. McKell[1], Courtney J. Campbell[3], Juan P. Barraza[1], Evan S. Krystofiak[4], Erin R. Green[1,5], Tae Akizuki[6], Walter J. Chazin[6], Michael S. VanNieuwenhze[2], Timothy L. Stemmler[3], David P. Giedroc[2,7]*, Eric P. Skaar[1]*

1 Department of Pathology, Microbiology, and Immunology, and Vanderbilt Institute for Infection, Immunology, and Inflammation, Vanderbilt University Medical Center, Nashville, Tennessee, United States of America, 2 Department of Chemistry, Indiana University, Bloomington, Indiana, United States of America, 3 Department of Pharmaceutical Sciences, Wayne State University, Detroit, Michigan, United States of America, 4 Department of Cell and Developmental Biology and Cell Imaging Shared Resource, Vanderbilt University, Nashville, Tennessee, United States of America, 5 Department of Microbiology, University of Chicago, Chicago, Illinois, United States of America, 6 Departments of Biochemistry and Chemistry, and Center for Structural Biology, Vanderbilt University, Nashville, Tennessee, United States of America, 7 Department of Molecular and Cellular Biochemistry, Indiana University, Bloomington, Indiana, United States of America

☯ Authors contributed equally to this work.
* eric.skaar@vumc.org (EPS); giedroc@iu.edu (DPG)

## Abstract

To colonize and survive in the host, bacterial pathogens like *Acinetobacter baumannii* must acquire zinc (Zn). To maintain Zn homeostasis, *A. baumannii* synthesizes proteins of the COG0523 family which are predicted to chaperone Zn to metalloproteins. Bioinformatic tools identified *A. baumannii* A1S_0934 as a COG0523 protein, and yeast two-hybrid screening revealed that MurD, an essential muramyl ligase, interacts with A1S_0934. As such, we have named A1S_0934 MurD interacting GTPase COG0523 (MigC). Here we show that MigC is a GTPase whose activity is stimulated upon Zn coordination to a characteristic CxCC (C = Cys; x = Leu/Ile/Met) motif to form a $S_3$(N/O) complex. MigC-deficient strains (Δ*migC*) display sensitivity to Zn depletion and exhibit altered cell wall architecture *in vitro*. Biochemical and functional assays confirm the MigC-MurD interaction, which inhibits the catalytic activity of MurD. CRISPRi knockdowns of *murD* reduce *A. baumannii* fitness and increase filamentation during Zn depletion, a phenotype reversed in Δ*migC* strains, suggesting that MigC also inhibits MurD activity in cells. Δ*migC* cells are elongated and sensitized to ceftriaxone, a cephalosporin antibiotic, consistent with decreased cell wall integrity. The Δ*migC* strain has reduced ability to colonize in a murine model of pneumonia highlighting the importance of the MigC-MurD interaction induced by *A. baumannii* infection. Together these data suggest that MigC impacts cell wall biogenesis, in part through interactions with MurD, emphasizing the importance of MigC and MurD to the

**Data availability statement:** All relevant data and its supporting information files have been made publicly available on figshare at the following link: https://figshare.com/projects/The_Zinc_Metalloprotein_MigC_Impacts_Cell_Wall_Biogenesis_through_Interactions_with_an_Essential_Mur_Ligase_in_Acinetobacter_baumannii/238301.

**Funding:** The work presented here was supported by NIH R01 AI101171 (E.P.S., D.P.G., W.J.C.), R35 GM118157 (D.P.G.), T32 HL094296 (E.R.G), NIH F31AI169971 (J.M.C) and NIH T32 ES007028 (J.M.C). The funders had no role in study design, data collection and analysis, decision to publish, or preparation of the manuscript.

**Competing interests:** The authors have declared that no competing interests exist.

survival and pathogenicity of *A. baumannii* while expanding the potential functions of the COG0523 family of enzymes.

## Author summary

P-loop GTPase proteins of the COG0523 family are linked to adaptive growth during Zn deficiency. Here we demonstrate that A1S_0934, designated as MurD interacting GTPase COG0523 (MigC), modulates cell wall biogenesis and integrity through interactions that reduce the activity of MurD, an essential enzyme in the formation of peptidoglycan. Identification of MigC as a COG0523 protein and unveiling MurD as a binding partner of MigC underscores the importance of COG0523 proteins in bacterial physiology. Elucidating these interactions offers crucial insights into potential therapeutic strategies against multidrug-resistant *A. baumannii*, specifically by targeting essential cell wall enzymes such as MurD. These findings establish the potential of MigC as a novel drug target to disrupt vital pathways in peptidoglycan biosynthesis.

## Introduction

*Acinetobacter baumannii*, a gram-negative opportunistic pathogen, has emerged as a significant cause of hospital-acquired infections. Its rise to prominence is largely attributed to intrinsic and acquired resistance to multiple antibiotics, which complicates treatment and significantly increases the likelihood of therapeutic failure [1–4]. Recognizing the urgent need for new treatment options, the World Health Organization has designated *A. baumannii* as a "Priority 1: Critical Pathogen" [5]. Efforts to identify therapeutic strategies have largely focused on targeting essential bacterial processes, with cell envelope inhibitors, such as β-lactam antibiotics, being widely used in clinical settings [6,7]. However, the clinical utility of β-lactam antibiotics against *A. baumannii* infections has been significantly compromised due to widespread resistance, even when used alongside β-lactamase inhibitors. This limitation underscores the urgent need for innovative approaches that extend beyond simply enhancing β-lactam efficacy to exploit this crucial drug target more effectively.

The bacterial cell envelope interfaces with the extracellular environment and protects pathogens and commensals from environmental onslaughts, including the vertebrate immune system [8–15]. The gram-negative cell envelope is a three-layered structure composed of a bipartite membrane and a periplasmic cell wall constructed from a peptidoglycan scaffold. Peptidoglycan is composed of repeating N-acetylmuramic acid (MurNAc)-N-acetylglucosamine (GlcNAc) disaccharides linked by β1–4 glycosidic bonds, forming linear glycan strands [9–15]. Each MurNAc unit is further elaborated by a five-amino acid stem peptide, composed of *L*-alanine, *D*-glutamic acid, *meso*-diaminopimelic acid, and two terminal D-alanine residues in *A. baumannii* [9–15]. The composition and nature of crosslinking of the pentapeptide chain impacts the rigidity and physical integrity of the bacterial cell wall. The

MurNAc-peptide-GlcNAc lipid-linked peptidoglycan precursor is transported across the inner membrane to the periplasm polymerizing into the three-dimensional peptidoglycan structure. Peptidoglycan biosynthesis involves approximately 20 reactions, including cytoplasmic UDP-MurNAc-pentapeptide chain synthesis carried out by Mur ligases, making it a commonly targetable process for antibiotic development [6,7].

Mur ligases are a family of ATP-dependent enzymes that function early in *de novo* peptidoglycan biosynthesis, catalyzing the sequential addition of amino acids onto the D-lactoyl group of a UDP-MurNAc precursor, the latter formed by the combined action of MurA and MurB, which transforms UDP-GlcNAc to UDP-MurNAc [16–19]. The Mur ligases catalyze the formation of an amide bond with the simultaneous hydrolysis of ATP via an acylphosphate intermediate. Two divalent cations are required for activity, typically magnesium (Mg) or manganese (Mn), one for ATP binding (Mg) and another that stabilizes the acylphosphate intermediate (Mg or Mn), which is coordinated by a carbamoylated lysine residue. The Mur ligases are MurC, MurD, MurE, and MurF, and each possesses distinct substrate specificities. MurC catalyzes the addition of *L*-alanine to UDP-MurNAc, forming UDP-MurNAc-*L*-alanine (UMA), while MurD incorporates *D*-glutamic acid onto UMA, producing UDP-MurNAc-*L*-alanine-*D*-glutamate (UMAG) [16–19]. Mismetallation can impact the function and regulation of Mur ligases, presenting opportunities to therapeutically target metal homeostasis and peptidoglycan synthesis pathways [16–19].

Transition metals are critical micronutrients across all domains of life, underpinning a myriad of essential cellular processes. Zinc (Zn) is grouped with the late 3$d$-block transition metals and serves as a catalytic and structural cofactor for diverse proteins [20]. In vertebrates, stringent regulation of extracellular Zn levels, often by the immunity-associated protein calprotectin, allows the host to create environments that are either Zn-toxic or Zn-deficient, thereby challenging pathogens like *A. baumannii* to maintain mechanisms to adapt to fluctuations in Zn availability [10,12,21–24]. To maintain the functional integrity of Zn-dependent metalloproteins, Zn uptake and distribution within the cell are tightly regulated [24]. Part of this hierarchical distribution is thought to be mediated in part by specialized intracellular transport proteins known as metallochaperones, which deliver Zn to metalloproteins, ensuring optimal protein function and cellular homeostasis during metal restriction [24–27].

COG0523 proteins, which belong to the highly conserved G3E family of GTPases, encompass both established eukaryotic Zn metallochaperones like ZNG1 and putative bacterial metallochaperones like ZigA and ZagA [27–32]. These proteins typically harbor a CxCC metal-binding motif, demonstrated to bind Zn with high affinity in *A. baumannii* ZigA, along with a canonical N-terminal P-loop GTPase domain which likely acts in concert with the CxCC motif to facilitate the transfer of metal ions to target metalloproteins [29,31]. Despite the broad phylogenetic conservation of COG0523 proteins, the functions of most bacterial COG0523 proteins remain largely uncharacterized. We hypothesized that some COG0523 proteins may indeed serve other roles beyond metallochaperone activity [31,33]. Addressing this significant knowledge gap could provide critical insights into the broader functions of COG0523 proteins in bacterial Zn homeostasis.

This work establishes MigC (A1S_0934) as a Zn-binding COG0523 protein in *A. baumannii* with core features similar to ZigA [29,31,32]. Using a yeast-two-hybrid screen, we discovered that MurD, an essential enzyme in peptidoglycan synthesis, interacts with A1S_0934. We confirmed that A1S_0934 interacts with and inhibits MurD activity biochemically, leading to its designation as <u>M</u>urD-<u>i</u>nteracting <u>G</u>TPase <u>C</u>OG0523, or MigC. MigC coordinates Zn in the CxCC motif to form a four-coordinate $S_3$(N/O) complex which strongly stimulates its GTPase activity. This motif is important for *A. baumannii* to grow in metal-deficient environments. Moreover, MigC alters the antibiotic resistance, virulence, and morphological plasticity of *A. baumannii*, highlighting the roles of MigC and MurD in disrupting a vital cellular process. These findings emphasize the critical role of MigC in supporting *A. baumannii* growth and colonization under Zn-limiting conditions, particularly through interactions with MurD.

## Results

### *A. baumannii* A1S_0934 is a COG0523 protein with conserved metal-binding and GTPase motifs

Members of the G3E family of P-loop GTPases, including COG0523 proteins, exhibit functional diversity. To probe the conservation of *A. baumannii* A1S_0934, a targeted amino acid sequence analysis was performed with functionally related

homologs identified in a published Sequence Similarity Network analysis (SSN) [30] [Fig 1A]. Homologs of A1S_0934 were identified in a single SSN cluster (cluster 26), comprising 33 proteins exclusive to *Acinetobacter* spp. The genes that encode these proteins are not predicted to be regulated by the Zn-responsive transcription regulator Zur, despite sequence and predicted functional similarity to the Zur-regulated COG0523 protein, ZigA [23,31]. This exclusivity to *Acinetobacter* spp., coupled with the absence of regulation by Zur, suggests that A1S_0934 might represent a lineage-specific adaptation, warranting its prioritization for further study.

The distinct origin of cluster 26 in *Acinetobacter* species suggests a conserved protein function, and the CxCC metal-binding motif typical of the COG0523 protein family suggests a potential role as a metallochaperone [29–32]. The alignment of amino acids was performed using ClustalX [Fig 1B, 1C, and 1D] [33]. This analysis revealed that key motifs associated with protein function are present across all homologs of A1S_0934 identified in cluster 26 which were then visualized using Gecos CLI [34]. Specifically, it identified the Walker A (G1; P-loop) domain (GxxGxGK) which coordinates the β-phosphate, two "switch" loops, G2 (switch 1) and G3 (Walker B; switch 2) that sense G-nucleotide status (GTP vs. GDP), in between which is a cysteine-rich motif (CxCC). An AlphaFold3 model of A1S_0934 [S1 Fig] reveals an expected

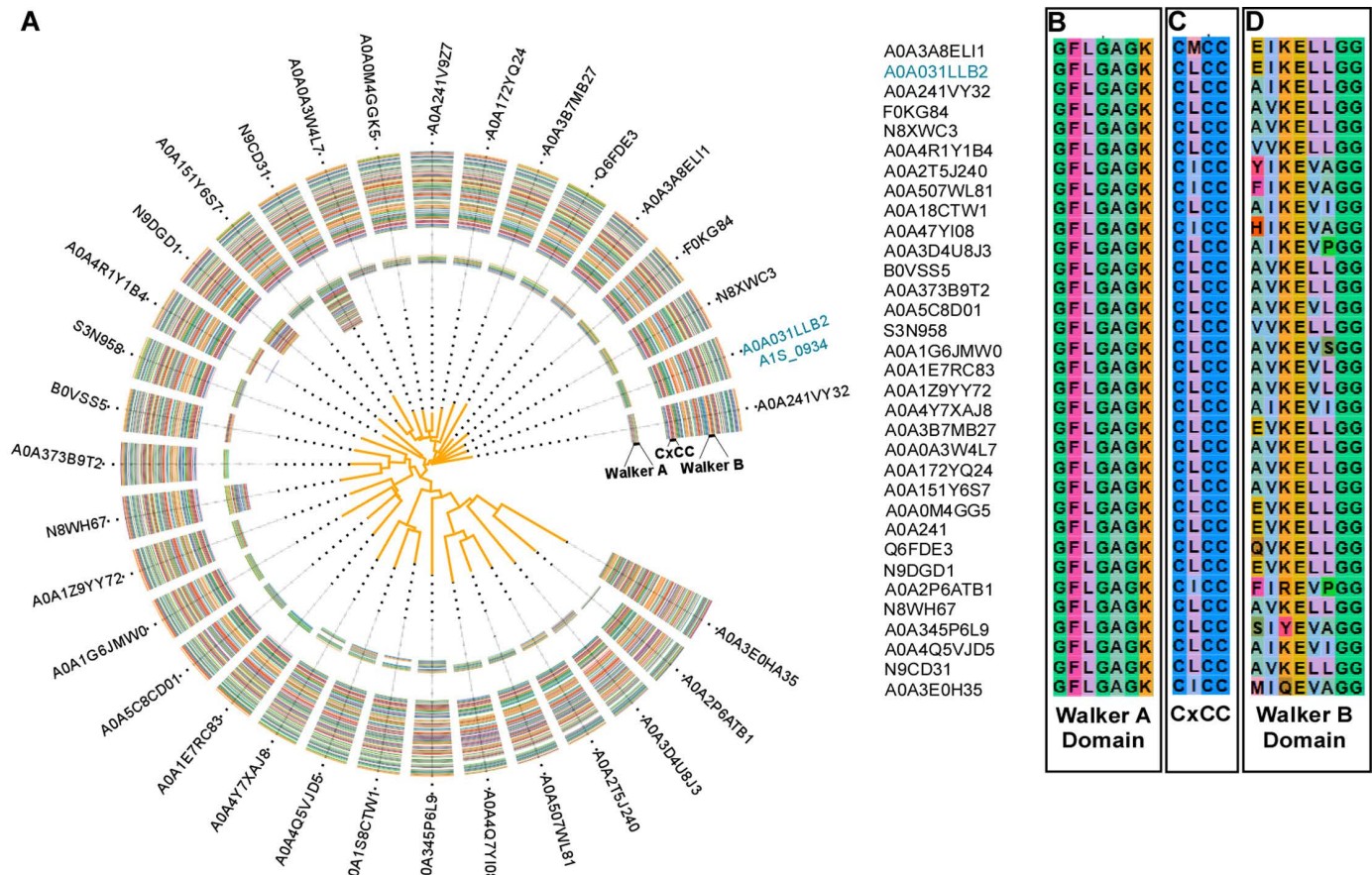

**Fig 1. A. baumannii A1S_0934 is a COG0523 protein with conserved metal-binding and GTPase motifs.** (A) Phylogram ClustalX analysis of A1S_0934 (MigC) amino acid sequence conservation from 33 homologs identified in a previously published Sequence Similarity Network analysis[30]. MigC is represented by the UniProt ID A0A031LLB2. Multiple sequence alignment radiates outward from amino acids 1-200, indicating the amino acids from the GTPase/CobW domain. Gecos visualization by pairwise comparison of the (B) Walker A, (C) CxCC, and (D) Walker B domains of the SSN-identified homologs of A1S_0934 [30].

N-terminal G-domain, harboring all five conserved G-loops that are responsible for GTP hydrolysis and metal binding, and a variable C-terminal domain thought to recruit interacting partner proteins [35]. Collectively, the analysis of A1S_0934 homologs highlight the conservation of key functional motifs and suggests a conserved role for A1S_0934 homologs across different species of *Acinetobacter*.

## A1S_0934 exhibits Zn-stimulated GTPase activity

The sequence homology of A1S_0934 to Zn-dependent COG0523 proteins suggests that A1S_0934 binds and utilizes Zn for its function. To assess this, metal competition was assayed with two Zn chelators that vary in their Zn-binding affinities, mag-fura-2 (MF2; $K_{Zn} = 5.0 \times 10^7$ M$^{-1}$) and quin-2 ($K_{Zn} = 2.0 \times 10^{11}$ M$^{-1}$) [29]. A titration of Zn into a solution of the low affinity competitor, MF2, revealed that A1S_0934 binds two mol equivalents of Zn [Fig 2A]. The high-affinity metal binding site of A1S_0934 outcompetes MF2 ($K_{Zn1} \geq 10^9$ M$^{-1}$), preventing quantification of its binding affinity. The affinity of the low-affinity binding site was found to be $K_{Zn2} = 6.9$ (± 0.2) $\times 10^6$ M$^{-1}$. A titration of Zn into a solution of quin-2 and A1S_0934 revealed a single high-affinity metal binding site, characterized by $K_{Zn1} = 7.0$ (±0.5) $\times 10^{10}$ M$^{-1}$ [Fig 2B]. We hypothesized that, like *A. baumannii* ZigA, Zn and nucleotide binding are thermodynamically coupled in A1S_0934 [29,31]. Indeed, $K_{Zn1}$ is 20-fold larger, 1.5 (±0.1) $\times 10^{12}$ M$^{-1}$, for the A1S_0934-GDP complex [Fig 2C] and $K_{Zn1}$ is ≈ 40 times larger, 2.7 (±0.4) $\times 10^{12}$ M$^{-1}$, for the A1S_0934-GTP complex [Fig 2D]. This finding is consistent with the GTP-dependent metallochaperone model where nucleotide and metal occupancy stabilize an active conformation prior to client protein binding [32]).

Like all other studied COG0523 proteins, A1S_0934 has an invariant metal-coordinating cysteine (CxCC) motif, initially characterized in *A. baumannii* ZigA, that is likely the high-affinity Zn binding site. To probe the coordination structure of the high-affinity Zn binding site, X-ray absorption spectroscopy (XAS) of Zn1-bound A1S_0934 in either sodium chloride (NaCl)- or sodium bromide (NaBr)-containing buffers were carried out. Bromide ions scatter similarly to sulfur-containing ligands due to their size. In coordinatively unsaturated metal binding sites where a solvent molecule is a ligand, bromide and chloride ions would be expected to exhibit distinct scattering effects. However, the X-ray absorption near-edge structures (XANES) of each sample were identical. They revealed a $d^{10}$ electron configuration of Zn [Fig 2E]. Moreover, extended X-ray absorption fine structure (EXAFS) for both samples was best fit to a coordinately saturated tetrahedral $S_3$(O/N) coordination complex [Fig 2F, 2G, 2H, and 2I and S1 Table]. These data do not conclusively distinguish between oxygen (O) and nitrogen (N) ligation by the non-thiolate Zn ligand.

The GTP hydrolysis activity of A1S_0934 and its dependence on metal coordination by the high affinity Zn1 site were then evaluated. Apo-0934 exhibited minimal GTP hydrolysis activity, but the addition of 1 mol equivalent of Zn significantly enhanced this activity [Fig 2J]. Conversely, GTPase activity was minimal in the presence of saturating Mn, indistinguishable from *apo*-0934. Inorganic phosphate release assays were also performed with ATP and UTP. These assays revealed that A1S_0934 preferentially hydrolyzed GTP over all the conditions tested, with ATP being a poorer substrate, as found previously for ZigA, and UTP showing negligible activity [Fig 2J-K] [31]. An A1S_0934 mutant harboring a Walker B (G3: switch 2) substitution, E99A, exhibited background GTPase activity and minimal to no Zn dependence [Fig 2L]. Together these data demonstrate that A1S_0934 binds Zn with high affinity at a site comprising all three cysteines from the conserved CxCC motif and a fourth non-thiolate protein-derived ligand, and that metal binding is thermodynamically coupled to GTP binding and hydrolysis.

## A1S_0934 is critical for *A. baumannii* to overcome Zn limitation

A1S_0934 is a paralog of *Ab*ZigA, which contributes to Zn homeostasis, and A1S_0934 binds Zn [23]. Therefore, the contribution of A1S_0934 to the survival of *A. baumannii* during Zn deficiency was assessed. To achieve this, we generated an *A1S_0934* deletion strain (Δ0934), a complementation strain where a full-length copy of *0934* was introduced into Δ0934 at the mini-Tn7 site driven by the native promoter of *A1S_0934*, as well as empty integration controls (IC) or controls which contain an empty vector insertion at the mini Tn7 site (both in wild-type (WT) and Δ0934) (S2 Table). WT

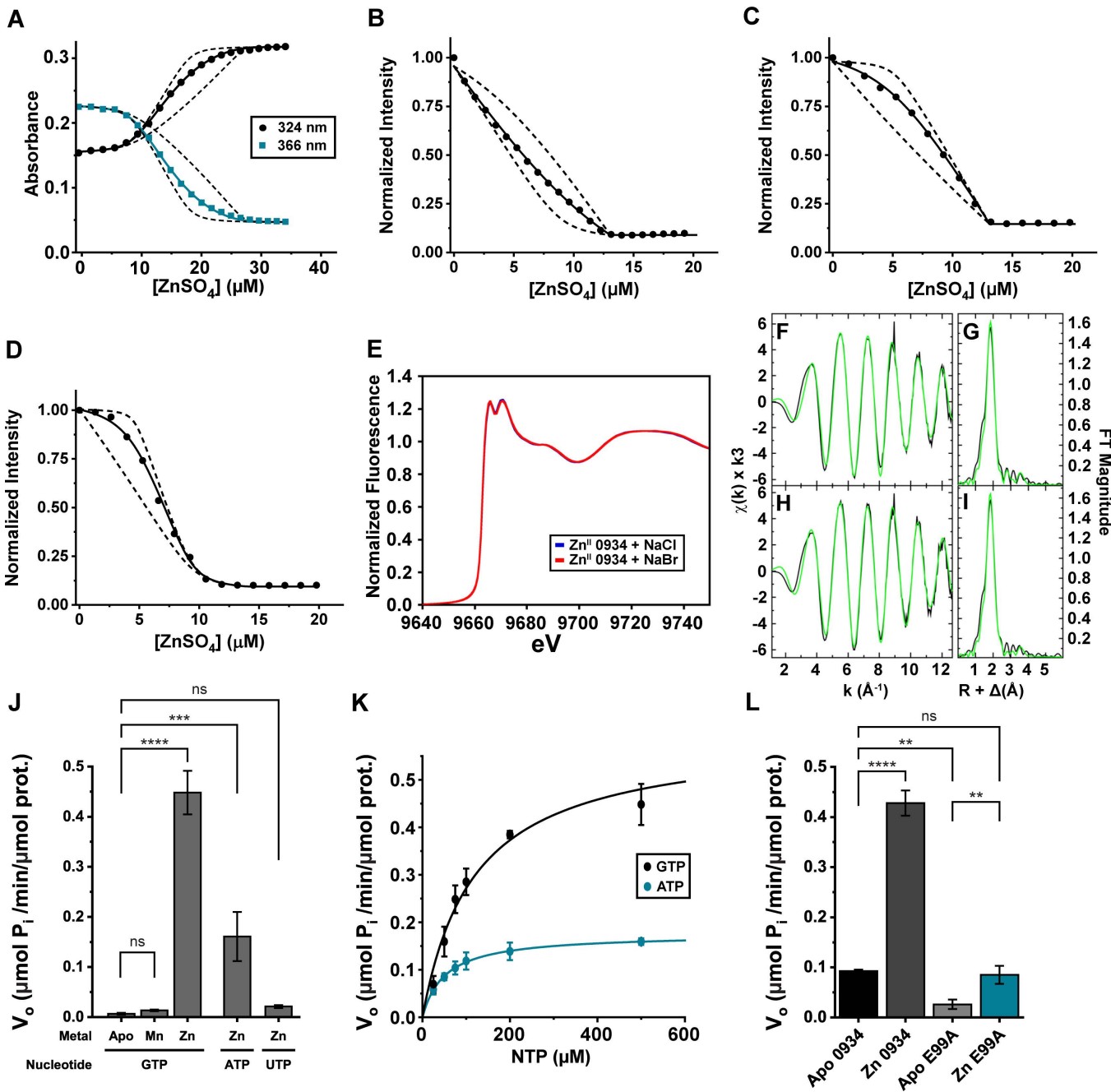

**Fig 2. A1S_0934 is a COG0523 Zn-binding GTPase.** (**A**) Competitive metal binding titration with A1S_0934 and mag-fura-2 (MF2) quantifies the second binding site of A1S_0934 for $Zn^{2+}$ measuring absorbance in the UV-visible spectrum at 324 nm and 366 nm. Representative titration of Zn into 9.09 μM A1S_0934 and 9.48 μM MF2. Curve fitting (solid line) determined A1S_0934 $K_{Zn2}$ = 6.9 (± 0.2) × $10^6$ $M^{-1}$. Dashed lines represent simulations with 10-fold lower and 10-fold higher binding affinities to illustrate the robustness of curve fitting. (**B-D**) Representative competitive metal binding titrations with A1S_0934 and quin-2 quantify the high affinity binding site of A1S_0934 under various conditions by fluorescence intensity at 490 nm. Curve fitting (solid line) determined A1S_0934 without nucleotide to have $K_{Zn1}$ = 7.0 (± 0.5) × $10^{10}$ $M^{-1}$, A1S_0934 in the presence of GDP to have $K_{zn1}$ = 1.5 (±0.1) × $10^{12}$ $M^{-1}$, and A1S_0934 in the presence of GTP to have $K_{Zn1}$ = 2.7 (±0.4) × $10^{12}$ $M^{-1}$. Titrations of A1S_0934 in the presence of GTP fit to a two-state binding model with $K_{zn2}$ = 5.7 (±1.3) × $10^9$ $M^{-1}$. Dashed lines represent simulated models with 10-fold lower and 10-fold higher $K_{Zn1}$ binding affinities to illustrate the robustness of curve fitting. Representative titrations are shown of Zn titrated into (**B**) 4.37 μM A1S_0934 and 8.69 μM quin-2 (**C**) 5.21 μM A1S_0934, 7.89 μM quin-2, and 1 mM GDP (**D**) 4.64 μM A1S_0934, 4.65 μM quin-2, and 1 mM GTP. (**E**) Normalized XANES (X-ray absorption near

edge structure) for Zn$^{II}$-bound A1S_0934 in NaCl (blue) and NaBr (red). (**F-I**) Zn EXAFS (Extended X-ray absorption line structure) and Fourier transformation of A1S_0934 in NaCl (**F-G**) and NaBr (**H-I**). (**J**) Malachite green assay measuring rate of NTP (nucleotide-triphosphate) hydrolysis of 5 µM A1S_0934 in the presence of Zn (5 µM) or Mn (500 µM) as indicated. (**K**) Michaelis-Menten kinetics of 5 µM Zn bound A1S_0934 for ATP and GTP. Curve fitting (solid line) was performed to get enzyme kinetic parameters. Reactions with GTP calculated $V_{max}$ = 0.59 ± 0.06 µmol/min/µmol A1S_0934 and $K_m$ = 110 ± 14 µM GTP. Reactions with ATP calculated $V_{max}$ = 0.175 ± 0.006 µmol/min/µmol A1S_0934 and $K_m$ = 51.1 ± 12 µM ATP. (**L**) Malachite green assay measuring rate of GTP hydrolysis of WT A1S_0934 and Walker B mutant E99A A1S_0934 in the presence and absence of stoichiometric Zn. **$p<0.01$, ****$p<0.0001$ by one-way ANOVA.

and Δ*0934 A. baumannii* were exposed to the broad-spectrum metal chelator (*N,N,N',N'*–tetrakis(2-pyridinylmethyl)-1, 2-ethanediamine (TPEN), revealing that inactivating *A1S_0934* increases sensitivity to metal depletion, which is fully rescued by the expression of *A1S_0934 in cis* [Fig 3A] and upon ZnCl$_2$ supplementation [Fig 3B]. This suggests that A1S_0934 may be involved in managing cellular Zn scarcity. Expression of *A1S_0934 in cis* restored WT growth levels, however this requires an intact metal binding site (C71, C73, C74) and GTPase activity (E99), as mutants lacking these functional domains failed to fully recover growth Fig 3C]. This suggests that both metal-binding and GTPase activity are important for the function of A1S_0934 during chelator-induced cellular metal starvation.

Since *A. baumannii* experiences multi-metal starvation by the host defense protein calprotectin (CP), we examined the *in vitro* response of WT and Δ*0934* to treatment with CP harboring a His6 mutation (in metal binding site 2) which makes CP specific for Zn chelation [36–41]. Δ*0934* was particularly susceptible to CP-induced Zn sequestration and growth was fully restored by the expression of *A1S_0934 in cis* [Fig 3D]. Given the heightened susceptibility of Δ*0934* to Zn sequestration, we hypothesized that inactivating *A1S_0934* affects cell shape akin to the morphological phenotypes depicted in the Δ*zigA* strain [12,31]. To evaluate morphological and structural changes of *A. baumannii* during Zn starvation, transmission electron microscopy was performed on WT and Δ*0934* strains. WT *A. baumannii* cells are rounded in Zn-restricted medium while Δ*0934* cells are elongated [Fig 3E]. Further, Δ*0934* cells grown in TPEN exhibited increased cell diameter [Fig 3F], peptidoglycan [Fig 3G], cell envelope width [S2A Fig], and inner membrane width [S2B Fig] with no significant changes to the outer membrane width [S2C Fig].

Given these findings, a series of cell wall labeling studies were performed where newly synthesized peptidoglycan was labeled with the fluorescent *D*-amino acid, HADA, in the fourth position of the peptide stem, allowing the monitoring of WT and Δ*0934* peptidoglycan during metal starvation induced by TPEN [Fig 3H] [42]. A significant increase in cell volume was observed as well as a trend toward increased HADA signal in TPEN-treated Δ*0934* cells [Fig 3I-J]. These data suggest that A1S_0934 modulates *A. baumannii* survival by impacting cellular morphology, thus contributing to resistance against *in vitro* metal sequestration.

## A1S_0934 binds and inhibits the activity of the Mur ligase MurD

Considering the predicted role of COG0523 orthologs, such as *A. baumannii* ZigA, in facilitating growth during Zn deficiency by metal binding and transfer to a partner protein, we hypothesized that A1S_0934 operates in a similar manner [7,30]. To identify potential clients of A1S_0934, a yeast-two-hybrid (Y2H) screen was employed, revealing MurD as the sole high-confidence interactor, with the minimal interaction region mapping to amino acid residues 53–103 of MurD [Fig 4A] [43]. This interaction was further corroborated by a one-by-one Y2H assay, demonstrating that full-length A1S_0934 binds to a truncated MurD variant (AA 55–354), pinpointing critical amino acids involved in their binding [S3A Fig]. MurD, like other Mur ligases, is a three-domain enzyme, and residues 53–354 are fully coincident with domain 2 (d2) which harbors a MurD ligase catalytic site in MurD. We therefore designate A1S_0934 as <u>M</u>urD-<u>i</u>nteracting <u>G</u>TPase <u>C</u>OG0523 or MigC.

An HPLC-based assay was developed that leverages the ability to separate the MurD substrate, UDP-N-acetylmuramyl-*L*-alanine (UMA), from product, UDP-N-acetylmuramoyl-*L*-alanine-*D*-glutamate (UMAG), and implemented to evaluate the impact of MigC on the steady-state kinetics of MurD activity. These kinetics reactions were performed with saturating *D*-glutamate and

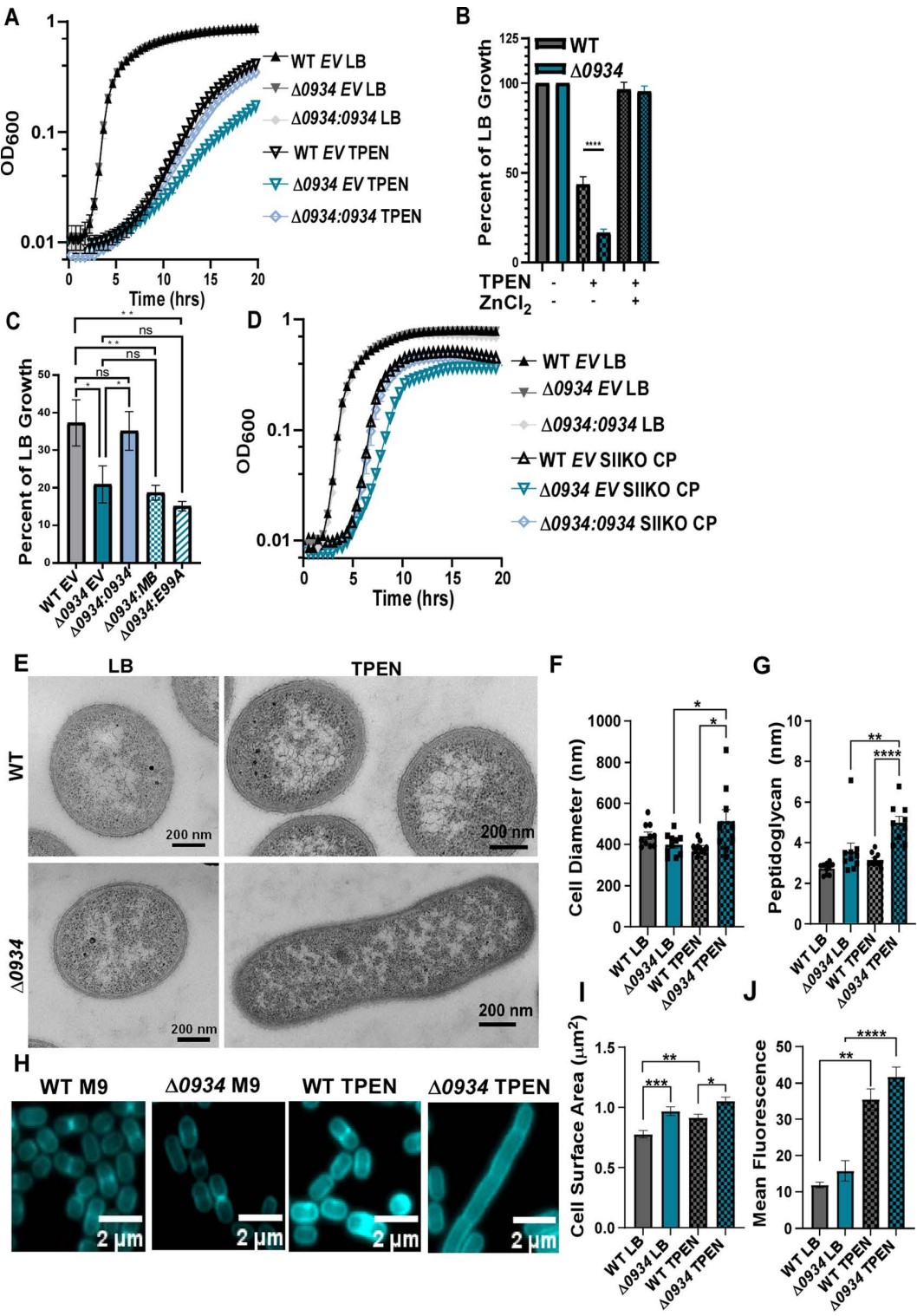

**Fig 3. A1S_0934 is required for full growth of *A. baumannii* during Zn-limitation. (A)** WT or Δ*0934* integration controls (IC) or Δ*0934::0934* were grown in LB ± 60 μM TPEN with $OD_{600}$ monitored over time. **(B)** Percentage of rich medium (LB) growth as determined by $OD_{600}$ at 15 h after growth in either 60 μM TPEN or 60 μM TPEN ± 0.5 μM $ZnCl_2$ compared with untreated strains. ****p < 0.0001 by one-way ANOVA. **(C)** Percentage of rich medium (LB) growth as determined by $OD_{600}$ at 15 h after growth in 60 μM TPEN of WT (IC), Δ*0934* (IC), Δ*0934::0934*, Δ*0934:: MB* (*C71A, C73A,* and *C74A*),

or Δ*0934::E99A* with OD$_{600}$ monitored over time. **(D)** WT or Δ*0934* containing an integration control (IC) or Δ*0934::0934* were grown ± 250 µg/ml His6 knockout (SIIKO) calprotectin and OD$_{600}$ was monitored over time. **(E)** Transmission Electron microscopy (TEM) of WT or Δ*0934* cells in LB ± 40 µM TPEN. Cells were further assessed for **(F)** cell diameter and **(G)** peptidoglycan width using ImageJ software. *p < 0.05, **p < 0.01, ****p < 0.0001 by one-way ANOVA. **(H)** Confocal imaging of WT or Δ*0934* grown in M9 minimal medium ± 10 µM TPEN with a HADA fluorescent amino acid probe. Cells were further assessed for **(I)** cell surface area and **(J)** mean fluorescence intensity using ImageJ software. *p < 0.05, **p < 0.01, ***p < 0.001, ****p < 0.0001 by one-way ANOVA.

UMA, and variable ATP. MurD was found to have a $V_{max} \approx 430$ µmol min$^{-1}$ µmol$^{-1}$ MurD with a $K_m$ for ATP of ≈150 µM [Fig 4B], consistent with literature values obtained for other bacterial MurD enzymes [44]. Zn-bound MigC reduces the $V_{max}$ of MurD to ≈230 µmol/min/µmol MurD with the $K_m$ unchanged, revealing that Zn-MigC is an allosteric or noncompetitive inhibitor of MurD [45]. Assays carried out over a range of Zn-MigC concentrations revealed an inhibition constant ($K_i$) of 32 ± 6 µM with an average of ≈75% inhibition of MurD activity at saturating Zn-MigC [Figs 4C and S3B]. Strikingly, the structurally similar MurC showed no MigC-dependent inhibition over the same range of Zn-MigC concentrations, revealing inhibition by MigC is specific to MurD [Fig 4C]. Inhibition rates determined from reactions containing *apo*-MigC without GTP, *apo*-MigC with GTP, and Zn-E99A MigC with GTP displayed levels of inhibition comparable to assays carried out with Zn-MigC with GTP [S3B Fig]. These data collectively reveal a modest affinity of the MigC-MurD complex that does not fully capture the functional requirements of MigC in cells

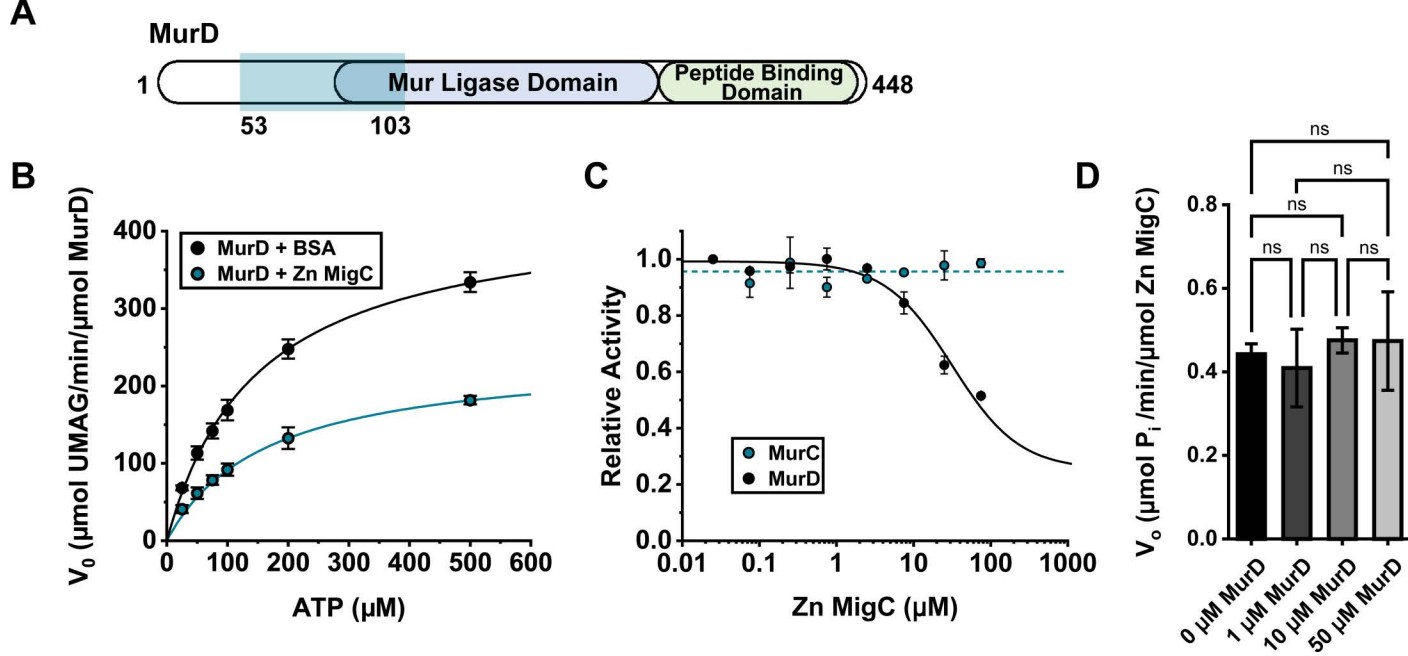

**Fig 4. MigC interacts with and decreases the activity of MurD. (A)** Yeast two-hybrid assay assessing interacting proteins with MigC. Teal shading indicates the minimal interaction domain of MurD with MigC. **(B)** Michaelis-Menten kinetics of 40 nM MurD with varied ATP concentrations measuring UMAG production by HPLC. Reactions were performed with bovine serum albumin (BSA) added (black) and with 4 µM Zn MigC added (blue). Curve fitting of reactions with added BSA calculated $V_{max}$ = 432 ± 31 µmol/min/µmol MurD and $K_m$ = 149 ± 25 µM ATP. Curve fitting of reactions with added Zn MigC calculated $V_{max}$ = 232 ± 16 µmol/min/µmol MurD and $K_m$ = 144 ± 31 µM. **(C)** Inhibition curves measuring the relative activity of MurC and MurD with various Zn MigC concentrations. MurC and MurD activity were quantified by measuring MurC product, UMA, and MurD product, UMAG, by HPLC. Curve fitting of MurD data (solid line) determined the $K_i$ for Zn MigC to be 32 ± 6 µM. A value for $K_i$ could not be determined for MurC data. Dotted line represents the average relative activity of MurC measurements. **(D)** Malachite green assay measuring the rate of GTP hydrolysis of 5 µM Zn MigC in the absence and presence of MurD at the indicated concentrations (*n* = 3). Significance was assessed by one-way ANOVA.

[Fig 2C]. In contrast, ligand-free MurD has no impact on the GTP hydrolysis activity of Zn-MigC under conditions where the two proteins interact [Fig 4D].

## MigC-MurD interactions alter *A. baumannii* morphological plasticity, antibiotic resistance, and virulence

MigC negatively impacts turnover of MurD *in vitro*, suggesting a model where a Δ*migC* strain may have increased cellular MurD activity. As *murD* is an essential gene, a CRISPRi system was applied in *A. baumannii* to genetically probe this interaction that makes use of anhydrotetracycline (AhTc)-inducible *dcas9* gene integrated at the *att*Tn7 site. To do this, we employed a constitutive *murD*-targeting single stranded guide RNA (sgRNA) scaffold that is maintained on pYDE007, a high-copy-number plasmid derived from pWH1266 [46–48]. Twenty-four-base pair *murD*-sgRNA targeting the nontemplate strand downstream of the predicted transcription start site was designed to silence *murD* [S3 Table], and then the plasmid was electroporated into WT and Δ*migC* containing an inducible copy of *dcas9* (referred to as WT or Δ*migC murD* KD for simplicity). In the absence of AhTc induction, there was no growth defect in the *murD* KD strains when grown in LB compared to the empty vector controls [S4A-C Fig]. However, when *dcas9* is induced with AhTc, both strains harboring the *murD*-targeting sgRNA exhibited growth delays that were dose-dependent and influenced the fitness of the empty vector controls significantly less [Figs 5A, 5B, S4A, S4B, and S4C]. This fitness defect appears delayed in Δ*migC*, consistent with an increase in cellular MurD activity, as anticipated by a loss of MigC-dependent MurD inhibition [Fig 5B]. This is further supported by maximum bacterial density calculations, showing that Δ*migC murD* KD exhibits a fitness advantage over WT when treated with 75 ng/mL AhTc (($OD_{600}$) 0.5430 vs. 0.4590)

Since peptidoglycan is an integral part of cell structural integrity, we hypothesized that genetic disruptions in *murD* would impact cellular morphology. With a sgRNA targeting *murD*, cells that were uninduced demonstrated coccobacilli morphology much like WT *A. baumannii* [Fig 5C], while induction of *dcas9* resulted in filamentation in both WT and Δ*migC* strains harboring a *murD*-targeting sgRNA [Fig 5C]. Filamentation in the Δ*migC murD* knockdown appears less severe, further indicating that this strain may be protected from the inactivation of *murD*. Growth [Fig 5D] and viability [Fig 5E] assessments of WT and Δ*migC* strains harboring a *murD*-targeting sgRNA showed that *dcas9* induction in Δ*migC* resulted in less severe effects. Growth rate analysis of Δ*migC* strains with *murD*-targeting sgRNA showed an increase in the area under the curve from 1.512 to 3.620 ($OD_{600}$ x Time (hrs.)) upon AhTc treatment, suggesting that reduced *murD* regulations mitigates some of the detrimental effects of disrupted cell wall biogenesis. We then further hypothesized that in low Zn conditions, the fitness burden in Δ*migC murD* knockdown strains might be further alleviated, since Zn may impact MurD activity or mediate a compensatory stress response that mitigates the effect of *murD* inactivation. To assess this, WT and Δ*migC* strains harboring a *murD*-targeting sgRNA were grown in the presence of the Zn chelator TPEN ±AhTc induction. Consistent with the hypothesis, it was found that fitness was improved in the Δ*migC* strain [Fig 5F].

Considering that disruptions in either *migC* or *murD* result in aberrant cell wall architecture, we hypothesized that *A. baumannii* lacking either of these genes is more sensitive to cell wall-targeting antibiotics, such as the β-lactam ceftriaxone (CRO). Indeed, Δ*migC* demonstrated reduced fitness compared to WT in CRO-treated medium, a deficiency that was fully restored by expression of *migC in cis* and was mediated by the metal binding and GTPase motifs of MigC [Figs 5G and S5A]. We then hypothesized that this compromise in cell wall integrity would impair the survival of *A. baumannii* within a vertebrate host due to increased vulnerability to antimicrobial pressures. To determine the contribution of MigC-MurD interactions *in vivo*, a murine pneumonia model of infection was performed and bacterial burdens from the lungs, liver, heart, kidney, and spleen were quantified. This experiment identified a significant attenuation in the ability of Δ*migC* to colonize the lungs [Fig 5H] and to disseminate to the heart [Fig 5I] and kidneys [Fig S5B], but not the liver [S5C or S5D Fig] spleen. These findings indicate that disruptions in *migC* and *murD* impair cell wall integrity and disrupt low Zn stress responses, sensitizing *A. baumannii* to CRO and potentially limiting its ability to colonize a vertebrate host. While the modulation of MurD activity by MigC likely contributes to these phenotypes, additional functions of MigC, particularly *in vivo*, may also play a significant role.

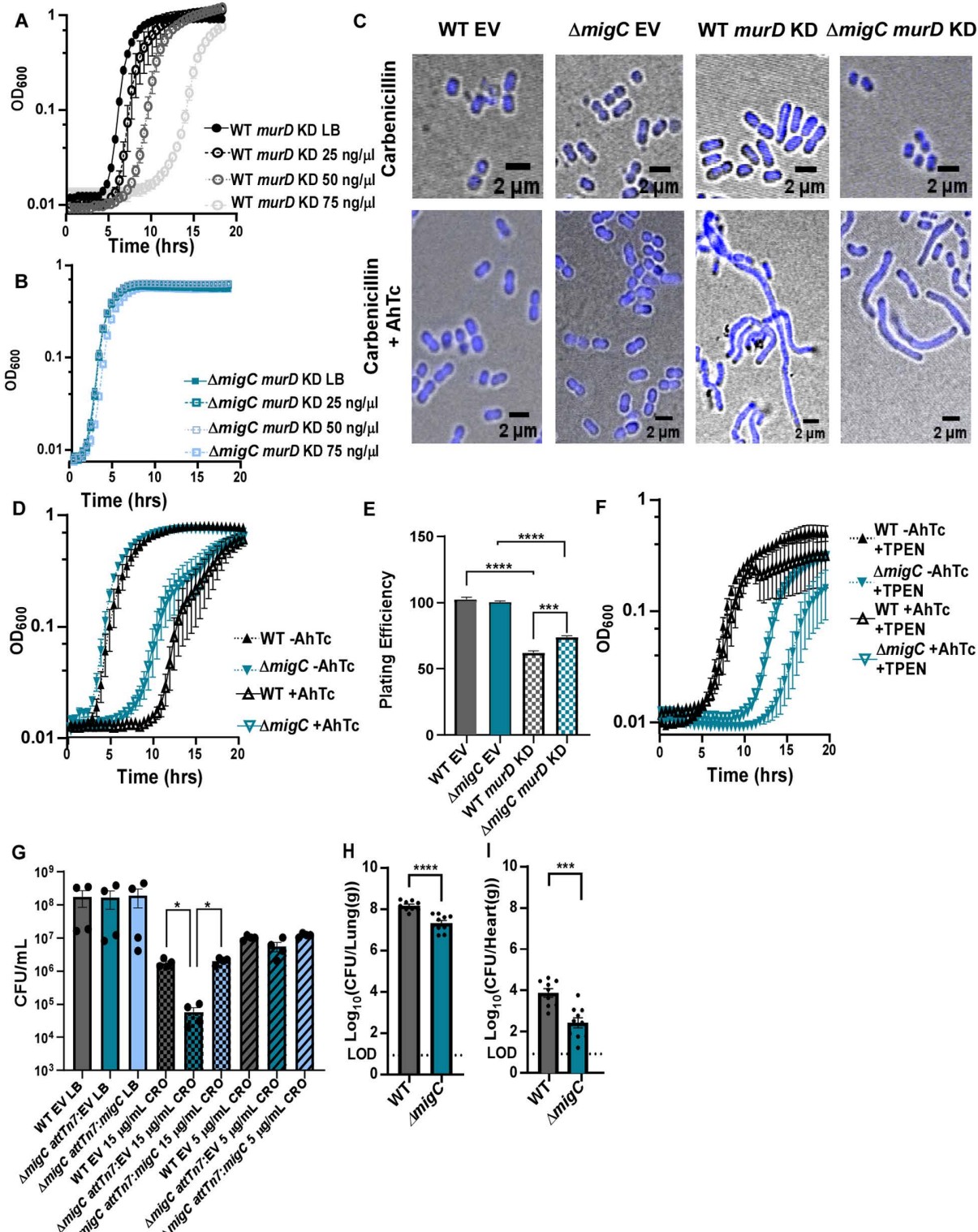

**Fig 5. Disrupting MigC-MurD interactions alters *A. baumannii* morphological plasticity, antibiotic resistance, and colonization. (A)** WT *murD* KD or **(B)** Δ*migC murD* KD were grown in LB with or without 25, 50, or 75 ng/mL anhydrotetracycline (AhTc) added. **(C)** WT *murD* KD and Δ*migC murD* KD were grown in ± 75 ng/mL AhTc and then stained with a DAPI DNA dye and imaged on a confocal microscope. DIC images were captured simultaneously and overlaid. **(D)** These same strains were grown ± 100 ng/mL AhTc with $OD_{600}$ monitored over time. **(E)** WT *murD* KD and Δ*migC murD* KD

were grown ± 75 ng/mL AhTc until mid-exponential phase and then were serial-diluted to plates containing 75 µg/mL carbenicillin to assess viability. ***p < 0.001, ****p < 0.0001 by one-way ANOVA. **(F)** These same strains were grown ± 50 ng/mL AhTc ± 20 µM TPEN with $OD_{600}$ monitored over time. **(G)** WT and Δ*migC* integration controls and the complementation strain were grown in LB ± either 15 µg/ml or 5 µg/ml of Ceftriaxone (CRO) and then plated to LBA and bacterial burdens were quantified. *p < 0.05 by one-way ANOVA. **(H-I)** Mice were intranasally infected with WT or Δ*migC*. Bacterial burdens were assessed at 36 hpi in the **(I)** lungs or **(J)** heart. Data are represented as the mean ± SEM with each point indicating the bacterial burdens from an individual mouse in a specific organ. The limit of detection is indicated as LOD. ***p < 0.001, ****p < 0.0001 by unpaired *t* test.

## Discussion

Zn is an essential micronutrient across all domains of life, playing a vital role in cellular processes such as cell wall biosynthesis. In Zn-deficient conditions, *A. baumannii* upregulates key cell wall linkages and outer membrane vesicle formation, indicating the importance of cell wall regulation in response to Zn limitation [10,49–51]. This study describes the discovery of *A. baumannii* MigC, a Zn-binding GTPase, by examining its interaction with MurD, a cytoplasmic enzyme essential for peptidoglycan synthesis. MigC reduces MurD activity, impacting cell wall synthesis and integrity. CRISPRi experiments revealed that silencing *murD* in Δ*migC* strains results in fewer growth and morphological defects compared to WT, suggesting compensatory regulation of MurD [52]. This is evident in Δ*migC* cells expressing a *murD* sgRNA, which exhibit reduced filamentation and improved growth, particularly during Zn limitation. Additionally, disruption of *migC* enhances β-lactam susceptibility and impairs survival in a murine pneumonia model, underscoring an important role of MigC-MurD interactions in maintaining cell wall integrity and bacterial fitness.

G3E P-loop GTPases, of which COG0523 enzymes are a subset, have been associated with metallochaperone activity due to their conserved metal-binding motifs and GTPase functionality [24,30]. This superfamily of proteins is often linked to metal ion trafficking and homeostasis [24–27]. However, emerging studies suggest that G3E GTPases may have more specialized functions beyond their anticipated role in metalloprotein maturation, likely exhibiting functional diversity across species and contexts [32,51]. Here we show that while MigC binds Zn and hydrolyzes GTP, we have no evidence that MigC delivers Zn to MurD, to the active site or elsewhere in a manner anticipated for a metallochaperone. Instead, MigC exerts a negative regulatory effect on MurD activity, with no obvious reciprocal impact on GTP hydrolysis by MigC [Fig 4B, 4C, and 4D]. This regulation modulates cell wall biosynthesis and consequently impacts bacterial morphology, indicating a specialized or perhaps moonlighting function for MigC in the control of cell wall dynamics rather than in metal ion trafficking. We hypothesize that inhibition of MurD by MigC fine-tunes flux through the peptidoglycan synthesis pathway, preventing precursor overaccumulation that could impact cell wall homeostasis. This regulatory role is reinforced by CRISPRi data showing that silencing *murD* in a Δ*migC* strain rescues the growth and morphological defects observed in WT cells [Fig 5C-F].

Although the Mur ligases- MurC, MurD, MurE, and MurF- share structural and functional similarities, our findings suggest that MigC selectively inhibits MurD [Fig 4C] [16–19]. This specificity is likely driven by as yet unidentified differences in structure or dynamics of the complex. The GTPase activity of MigC is highest in the Zn-bound state and the inhibition of MurD in cells is reliant upon the metal binding and nucleotide hydrolysis as indicated by the inability of metal binding and GTPase mutants to rescue cellular growth in Δ*migC* strains [Fig 3C]. Microscopic analysis reveals that disruption of the MigC-MurD interaction leads to cellular elongation. This elongated morphology is associated with increased susceptibility to antibiotics and reduced virulence, consistent with prior studies that link similar morphological phenotypes to *murD* inactivation in *Bacillus subtilis* and *Mycobacterium tuberculosis* [53,54]. We hypothesize that the elongated morphology observed in Δ*migC* cells is due to increased MurD activity, which disrupts the equilibrium of peptidoglycan synthesis and recycling, thereby affecting cell wall integrity. The morphological alterations induced by MigC-MurD interactions correlate with altered antibiotic resistance profiles, an observation supported by prior studies associating cell elongation with increased antibiotic susceptibility [55,56].

Moving forward, it will be critical to investigate the mechanistic underpinnings of these morphological changes, particularly how *migC* and *murD* intersect with other cellular pathways to influence antibiotic resistance and virulence. Close examination of MurD product profiles as resolved by our LC workflow reveals only the single expected product, UMAG, in the presence and absence of MigC, at saturating ATP and *D*-glutamate concentrations [S6 Fig]. This suggests that specific conversion of UMA to UMAG by MurD is unchanged by MigC *in vitro*. It remains formally possible that MigC influences MurD substrate specificity and product distribution in cells, while also modulating flux through this pathway. It also seems plausible that MigC interacts with other client proteins or pathways and such interactions may contribute to these phenotypes. Both possibilities are currently under investigation.

In summary, this study reveals that MigC, a Zn-binding COG0523 protein in *A. baumannii*, regulates MurD, a critical enzyme in the early steps of peptidoglycan biosynthesis, with broad implications for bacterial growth, morphology, antibiotic resistance, and virulence. These findings position MigC as a key mediator of an essential cellular process and underscore its potential as a target for therapeutic intervention aimed at disrupting cell wall biosynthesis and combating multidrug-resistant *A. baumannii* infections. This study also advances the functional characterization of COG0523 proteins, extending their known roles beyond metallochaperone function.

## Materials and methods

### Ethics statement

All mouse studies were conducted in accordance with the Public Health Service Policy on Humane Care and Use of Laboratory Animals and were approved by the Institutional Animal Care and Use Committee of Vanderbilt University Medical Center (protocol number M1900043-00), under the oversight of the NIH Office of Laboratory Animal Welfare (OLAW), United States.

### Strain generation

The strains and plasmids used in this study are listed in S2 Table, and the primers listed in S3 Table. Using allelic exchange, the Δ*migC* (Δ*migC::aphA* or Δ*0934*) mutant was made through amplifying 1,000 bp of DNA in both the 5' and 3' flanking region of *migC* using ATCC 17978 *A. baumannii* genomic DNA as a PCR template. The *aphA* kanamycin resistance gene was then amplified using the vector pUC18 as a template. These amplicons were then cloned into pFLp2 using HIFI assembly (New England Biolabs). Conjugation was then used to move the pFLp2 construct into wild-type (WT) ATCC 17978 *Acinetobacter baumannii* using tri-parental conjugation with the *Escherichia coli* HB101 strain containing the pRK2013 helper plasmid. Matings were plated to Luria-Bertani agar (LBA) containing 75 μg/mL carbenicillin and 15 μg/mL chloramphenicol to select for strains with an integrated plasmid. A sucrose selection was then performed on LBA containing 10% sucrose to select for clones with a resolved integrated plasmid, which were then plated to LBA containing 40 μg/mL kanamycin to screen for the loss of *migC* and replacement with *aphA*. This was then confirmed using multiple PCRs that read into and out of *aphA* and whole genome sequencing (Genewiz).

For the Δ*migC attTn7::migC*, Δ*migC attTn7::migC MB (C71A, C73A, C74A)*, Δ*migC attTn7::migC E99A,* and the Δ*migC attTn7::migC* complementation strain, *migC* and its native promoter were amplified out of WT ATCC 17978 *A. baumannii* and combined into pKNOCK-mTn7-Amp digested with BamHI and KpnI by HiFi assembly. The mini-Tn7 was introduced into Δ*migC* as previously described [57]. Δ*migC attTn7::E99A* was made using a QuickChange II site-directed mutagenesis (Agilent) kit where the mini-Tn7 containing *migC* and its native promoter were used as a template and amplified using primers encoding the indicated point mutant. The PCR product was then DpnI digested, transformed into *E. coli*, and confirmed by Sanger sequencing. The resultant construct was introduced into Δ*migC* by quad mating. The metal binding mutant, or Δ*migC attTn7::MB*, was constructed using a gBlock containing point mutations in *C71*, *C73*, and *C74* and the native promoter of *migC*. The gBlock is 1,024 bp in length and was combined into pKNOCK-mTn7-Amp and digested with BamHI and KpnI by HIFI assembly.

For the WT and Δ*migC* ATCC 17978 attTn7::tetR-P$_{tet}$-dcas9-rrnBT1-T7Te Gm$^r$+pYDE007(sgRNA$_{murD}$) and integration controls, construction began by using a mini-Tn7 element containing *dcas9* (pYDE009). The mini-Tn7 element containing *dcas9* was integrated into WT and Δ*migC* using quad mating. A plasmid for sgRNA expression in *A. baumannii* (pYDE007) was used as previously described [47]. The empty vector controls (WT and Δ*migC* ATCC 17978 attTn7::tetR-P$_{tet}$-dcas9-rrnBT1-T7Te Gm$^r$+pYDE007) were used as non-targeting controls in the CRISPRi experiments. The plasmid encoding an sgRNA for *murD* targeting CRISPRi was constructed by PCR amplification of the gRNA scaffold of pYDE007 using a forward *murD*-targeting 5' primer with SpeI at the 5' end, and a reverse primer with a 5' ApaI. PCR products were then digested with SpeI and ApaI, ligated, and introduced into WT and Δ*migC* containing *dcas9* using electroporation. It was ensured that the *murD* targeting sequence targeted the non-template strand and had a 12-nucleotide seed region found once in the genome [47,57,58].

## Mouse pneumonia model of infection with *A. baumannii*

C57BL/6 8-week-old female mice were purchased from Jackson Laboratories for infection models. Two days prior to infection, WT and Δ*migC* bacteria were struck to LBA for single colony isolation and grown overnight. The night before the infection, 2 mL Luria-Bertani broth (LB) cultures were inoculated with colony-purified isolates and were grown for 16 h at 37 °C in an incubator shaking at 180 rpm. Bacteria were harvested, washed in PBS, and prepared for infection at a final inoculum of $3 \times 10^8$ CFU in 40 µl of PBS. Bacterial innocula were confirmed via serial dilution prior to infection. At 36 h post-infection (hpi), mice were euthanized and CFUs enumerated from the lungs, heart, liver, kidney, and spleen. Gram per organ was determined by calculating the weight of each tube before and after harvesting organs to ensure specific quantification. All animal procedures were approved by the Institutional Animal Care and Use Committee (IACUC) at Vanderbilt University and conducted in accordance with institutional and federal guidelines. Animals were closely monitored for signs of distress, and humane endpoints were applied as needed. Procedures were designed to minimize pain and discomfort, using appropriate anesthesia and analgesia, with euthanasia conducted per IACUC-approved protocols.

## Fluorescent D-amino acid labeling

Overnight cultures of WT or Δ*migC A. baumannii* were diluted 1:50 into LB for 2 h and then subcultured into M9 minimal medium with 0.5% sodium succinate and Vishniac's trace metal mix ± 10 µM TPEN as previously reported [10,59]. HADA (7-hydroxycoumarin-3-carboxylic acid-amino-D-alanine) was synthesized as previously described [42,60,61] and was used as a final concentration of 1 mM. Cells were incubated at 37 °C for 2 h, fixed for 1 h in 70% ethanol at 0 °C, and then washed with PBS. Cells were then imaged with a Zeiss LSM 880 Confocal Laser Scanning microscope equipped with a 1.4 Plan Apo 63X oil objective. Zen software was used to acquire images, and all quantification of fluorescence intensity was performed using FIJI [62]. All data are pooled from 100 cells per treatment and genotype.

## Transmission electron microscopy

WT and Δ*migC* cells were grown overnight, diluted 1:50 into LB, and incubated at 37°C with shaking for 1 hour. Cells were then subcultured 1:100 into LB with or without 30 µM TPEN and grown to mid-log phase. Fixation was performed in 2.5% glutaraldehyde in 0.1 M cacodylate buffer at room temperature for 1 hour and at 4°C for 24 h. Fixed cells were embedded in 2% agar, equilibrated in 30% glycerol, plunge-frozen in liquid ethane, and freeze-substituted in 1.5% uranyl acetate at -80°C for 48 h. Samples were raised to -30°C, washed with methanol, and infiltrated with HM20 lowicryl, which was polymerized under UV light at -30°C for 24 h. Sections (70 nm) were cut using a Leica UC7 ultramicrotome and placed on 300 mesh copper grids. Samples were stained with 2% uranyl acetate and lead citrate. TEM imaging was conducted on

 

a Tecnai T12 transmission electron microscope at 100 keV, with quantitative analysis performed using FIJI ROI manager with at least 100 measurements per treatment and genotype.

### Confocal microscopy of *murD* knockdown strains

Overnight cultures of WT or Δ*migC A. baumannii* ± *murD* sgRNA were diluted 1:100 into LB containing 75 µg/mL carbenicillin for 1 hour and then subcultured at 1:50 for 3.5 h in LB containing 75 µg/mL carbenicillin ± 75 ng/mL AhTc. Cells were incubated at 37°C, fixed for 1 hour in 70% ethanol at 0°C, and then washed with PBS. Cells were mounted onto a slide with the addition of ProLong Gold Antifade Mountant with DNA Stain DAPI (Thermofisher). Cells were then imaged with a Zeiss LSM 880 Confocal Laser Scanning microscope equipped with a 1.4 Plan Apo 63X oil objective. Zen software was used to acquire images were adjusted using FIJI software [62].

### Anhtet Viability Assay for *murD* knockdown strains

Overnight cultures of WT or Δ*migC A. baumannii* ± *murD* sgRNA were diluted 1:100 into LB containing 75 µg/mL carbenicillin for 1 hour and then were subcultured at 1:50 for 3.5 h in LB containing 75 µg/mL carbenicillin ± 75 ng/mL AhTc. Cells were incubated at 37°C and serial dilutions onto LBA containing 75 µg/mL carbenicillin were then performed. Bacterial colonies were enumerated and quantified across 2 experiments and 6 biological replicates.

### Yeast-two-hybrid assay and analysis

The yeast-two-hybrid screen was performed by Hybrigenics using full length *A. baumannii* MigC. The prey library was constructed from a WT ATCC 17978 *A. baumannii* cDNA library in collaboration with Hybrigenics. Any interaction partners were scored based on the degree of confidence of the hit, and any experimental artifacts were excluded from analysis [Fig 4A]. A second 1-by-1 interaction domain mapping yeast two-hybrid was performed by Hybrigenics. Yeast growth on medium with histidine added indicates interactions between the activation domain (AD) and binding domain (BD) fusion proteins. This was specifically confirmed using full-length MigC and a truncated form of MurD, amino acids 55–354.

### Bacterial growth assays

Two days prior, all strains were struck on LBA and grown overnight. Single colonies were then used to start individual overnight cultures. Overnight cultures of the strains (WT, Δ*migC*, WT *attTn7::EV*, Δ*migC attTn7::EV*, Δ*migC attTn7::migC*, Δ*migC attTn7::migC*$^{E99A}$, Δ*migC attTn7::migC*$_{metal binding}$ *(MB)*, WT *attTn7::dcas9*+pYDE007-sgRNA$_{murD}$, Δ*migC attTn7:dcas9*+pYDE007-sgRNA$_{murD}$, WT *attTn7:dcas9*+pYDE007 (EV), Δ*migC attTn7:dcas9*+pYDE007 (EV), or *ATCC 17978 attTn7::tetR-P$_{tet-}$murD* Gm$^{r}$ (WT *murD* OE)) were subcultured 1:100 into LB for 1 hour. Back dilutions were inoculated 1:1000 into LB containing various experimental conditions: (tetrakis-(2-pyridylmethyl)ethylenedi-amine (TPEN) (Sigma), calprotectin (His6 knockout) (SIIKO), carbenicillin (Fisher), AhTc (Sigma), or ceftriaxone (Thermofisher) as indicated in the figure legends. Growth was monitored at OD$_{600}$ using an Epoch 2 or BioTek Synergy 2 microplate reader. For growth in experiments containing the calprotectin with a His6 mutation (SIIKO), bacteria were back diluted into LB containing 40 percent calprotectin buffer (100 mM NaCl, pH 7.5 20 mM Tris-HCl, 3 mM CaCl$_2$, and 5 mM β-mercaptoethanol) with calprotectin added to the indicated concentration. Recombinant human calprotectin was used for all *in vitro* assays and was purified as indicated in previous work [38,63].

### Ceftriaxone killing assay

Overnight cultures of WT *attTn7::EV*, Δ*migC attTn7::EV*, and Δ*migC attTn7::migC* were grown for 16 h at 37°C. Cultures were then back diluted into LB for 1 hour before subculturing into LB ± 15 µg/ml or 5 µg/ml ceftriaxone. Serial dilutions onto LBA were then performed at time points 0, 2, 4, and 6 h post inoculation. Bacterial colonies were enumerated and quantified across 2 experiments and 6 biological replicates.

## MigC expression and purification

Full length *A1S_0934* encoding MigC was amplified from *A. baumannii* ATCC 17978 genomic DNA using MigC_pHIS_F and MigC_pHIS_R and cloned into pHIS-Parallel1 expression vector at the NdeI restriction site [64]. The plasmid was transformed into *E. coli* BL21 (DE3) competent cells by heat shock for 45 s at 42 °C and inoculated on LB supplemented with 100 $\mu$g/mL ampicillin and grown at 37 °C overnight. Colonies were collected and grown at 37 °C in 1 L of LB medium supplemented with ampicillin until an OD of 0.6-0.8 was reached. Protein expression was induced with 1 mM isopropyl β-D-1-thiogalactopyranoside (IPTG) and cells were grown for 18 h at 18 °C. Cells were collected by centrifugation (4000 rpm, 4 °C, 20 min) and resuspended in lysis buffer (25 mM HEPES, 500 mM NaCl, 2 mM EDTA, 2 mM tris(2-carboxyethyl)phosphine (TCEP), pH 8.0) and sonicated for 15 min (3s on, 9s off, 65% power) in an ice bath. Insoluble protein and DNA were removed from the resulting lysate with addition of 10% polyethyleneimine (PEI) and centrifugation (10,000 rpm, 4 °C, 20 min). Recombinant MigC was treated with 40% (w/v) ammonium sulfate and stirred at 4 °C for 30 min and protein was collected through centrifugation (10,000 rpm, 4 °C, 20 min). The pellet was resuspended and dialyzed into buffer A (25 mM HEPES, 50 mM NaCl, 2 mM EDTA, 2 mM TCEP, pH 8.0). MigC was purified by injection onto a HiTrap Q FF anion exchange column and eluted with a 0%-100% gradient of buffer B (25 mM HEPES, 500 mM NaCl, 2 mM EDTA, 2 mM TCEP, pH 8.0). Fractions containing MigC were combined and purified by size exclusion chromatography (HiLoad 16/600 Superdex column). Fractions containing >95% purity of MigC were combined and buffer exchanged into "metal-free" buffer (25 mM HEPES, 150 mM NaCl, 2 mM TCEP, pH 8.0, Chelex-treated) to remove excess EDTA. Protein identity was confirmed by electrospray ionization mass spectrometry and protein stocks were confirmed to be metal-free by ICP-MS. Protein concentrations were measured by UV-Visible spectroscopy using $\varepsilon_{280}$, of 62,910 M$^{-1}$ cm$^{-1}$. Single point mutagenesis was performed on pHIS-Parallel1 expression vector containing *migC* using MigC_E99A_pHIS_F and MigC_E99A_pHIS_R to mutate residue E99 to Ala. E99A mutant MigC was purified identically to WT MigC.

## MurC and MurD cloning and purification

Full length genes encoding MurD (*A1S_0245*) and MurC (*A1S_3335*) were amplified from *A. baumannii* ATCC 17978 DNA using MurD_pHIS_F and MurD_pHIS_R and MurC_pHIS_F and MurC_pHIS_R, respectively. The resulting PCR products were inserted into pHIS-Parallel1 expression vector separately at the NcoI restriction site downstream from a TEV protease cleavable linker region and a 6xHis tag [65]. Each expression plasmid was transformed into *E. coli* BL21 (DE3) cells through heat shock and grown at 37 °C in 1 L of LB medium supplemented with 100 µg/mL ampicillin until an OD of 0.6-0.8 was reached. IPTG (1mM) was added, and cells were grown for 18 h at 18 °C.

Both MurC and MurD were purified similarly. Cells were collected by centrifugation, resuspended in buffer containing 25 mM HEPES at pH 8.0, 500 mM NaCl, 2 mM EDTA, and 2 mM TCEP and lysed through sonication. Cell debris was removed through centrifugation. The crude lysate containing 6xHis MurC/MurD was purified through affinity chromatography on a NiNTA column. The 6xHis tag was removed by 12 h incubation with TEV protease at 4 °C and cleaved 6xHis tag was removed by an additional round of affinity chromatography on the NiNTA column. This was followed by size exclusion chromatography (G200 16/60) and fractions containing >95% purity of MurC/MurD were combined. Purified MurC/MurD was buffer exchanged with metal-free buffer containing 25 mM HEPES at pH 8.0, 150 mM NaCl, and 2 mM TCEP. Protein identity was confirmed by electrospray ionization mass spectrometry and protein stocks were confirmed to be metal-free by ICP-MS.

## Metal competition assays

Mag-fura-2 (MF2) titrations were performed on MigC in metal-free buffer containing 25 mM HEPES at pH 8.0, 150 mM NaCl, 2 mM TCEP. A 200 µL sample of 10 µM protein and 10 µM of MF2 was prepared and an initial absorption spectrum confirmed the concentration of protein at 280 nm and unmetalated MF2 at 366 nm. ZnSO$_4$ was added in 2 µM increments and the sample was left to equilibrate for 2 min prior to absorbance measurements. This process of Zn addition was

repeated until no concentration change of metalated MF2 was observed by the absorbance at 324 nm. DynaFit was used to implement a nonlinear least square fitting of the binding data to a simple competition model using the affinity of MF2 ($K_{Zn}$ = 5.0 × 10$^7$ M$^{-1}$) as described previously [65,66].

Quin-2 titrations were performed on MigC in metal-free buffer containing 25 mM HEPES at pH 7.4, 150 mM NaCl, and 2 mM TCEP. In titrations performed in the presence of nucleotide, 2 mM MgCl$_2$ and 1 mM GDP or GTP was added to the sample. A 3 mL sample of 5–10 μM MigC and 5–10 μM of quin-2 was made from aliquots thawed on ice and added to a cuvette without light exposure. An initial absorption spectrum confirmed the concentration of MigC at 280 nm and quin-2 at 261 nm ($\varepsilon_{260}$ = 37,500 M$^{-1}$ cm$^{-1}$). ZnSO$_4$ was added in 1 μM increments and the sample was left to equilibrate for 10 minutes between measurements. The sample was excited at 339 nm and measurements of fluorescence emission were taken at 490 nm. This process of Zn addition was repeated until there was no change in fluorescent emission at 490 nm. Fluorescence intensities were normalized to fluorescence intensity prior to Zn addition. Zn binding affinities of MigC were determined by DynaFit using global nonlinear least squares fit of two independent titrations with different ratios of protein and quin-2 concentrations using the known affinity of quin-2 for Zn ($K_{Zn}$ = 2.0 × 10$^{11}$ M$^{-1}$) [65,66].

## X-ray Absorption spectroscopy

Purified MigC was diluted to 150 μM in "metal-free" buffer (25 mM HEPES, 150 mM NaCl (or NaBr), 2 mM TCEP, pH 8.0, Chelex-treated), 0.95 mol equivalents of ZnSO$_4$ were loaded into MigC, and excess ZnSO$_4$ was removed through buffer exchange. Metal loading was determined to be >90% by ICP-MS. Metalated MigC was concentrated to 1 mM in a 10 kDa MWCO spin concentrator and protein concentration were confirmed by UV-Visible spectroscopy. 30% (v/v) pure glycerol was added to the final samples. These samples were loaded into Kapton-wrapped leucite XAS sample cells, flash frozen in liquid nitrogen, and stored at 77 K until beam exposure. Zn XAS was collected at the Stanford Synchrotron Radiation Light source (SSRL) on beamline 7–3. This beamline is equipped with a Si[220] double crystal monochromator with a mirror upstream to the cryostat for beam focusing and harmonic rejection. Fluorescence excitation spectra were collected using a Canberra 30 element germanium solid state detector. The temperature during collection was maintained at 10 K using an Oxford Instruments continuous-flow liquid helium cryostat. XAS spectra were collected in 5 eV increments in the pre-edge region, 0.25 eV increments in the edge region, and 0.05 Å$^{-1}$ in the EXAFS region to $k$ = 14 Å$^{-1}$, integrated from 1 to 25 s in a $k^3$-weighted manner for a total scan length of ~40 min.

XAS spectra were processed and analyzed using the EXAFSPAK program suite written for Macintosh OS X and integrated with the Feff v8 software for theoretical model generation [66,67]. Normalized XANES spectra was subjected to edge analysis to determine the metal oxidation state [68]. EXAFS data, collected to $k$ = 14 Å$^{-1}$, were used for characterization of metal–ligand metrical parameters; data collected to this $k$ range correspond to a spectral resolution of 0.121 Å$^{-1}$ [69]; therefore, only independent scattering environments at distances >0.121 Å were considered resolvable in the EXAFS fitting analysis. Spectral simulations were performed using both single and multiple scattering model amplitudes and phase functions to deconvolute Zn–O/N, Zn–S, and Zn–C ligand interactions. For calibration of the theoretical models, a scale factor (Sc) of 1.0 and a threshold shift ($\Delta E_0$) of −15.25 eV (Zn–O/N/S/C) obtained from fitting crystallographically characterized small molecules, were used during simulations. The best-fit EXAFS simulations were based on the lowest mean square deviation between data and fit with, correction for the number of degrees of freedom ($F'$) [70]. During the standard criteria simulations, only the bond length and Debye–Waller factor was allowed to vary in each ligand environment.

## Malachite green assays

Reactions measuring MigC NTPase activity were performed in 25 mM HEPES, 150 mM KCl, 2 mM MgCl$_2$, 2 mM TCEP, pH 7.4. Zn containing MigC was pre-loaded prior to assays with 1 mol equivalent of ZnSO$_4$. Nucleotide-triphosphate (NTP) stocks of GTP (Jena Bioscience), ATP (Jena Bioscience), and UTP (Sigma Aldrich) were prepared in water, adjusted to

pH 7.0 and the concentration was confirmed with UV-Visible spectroscopy prior to experiments. Reactions (100 μL) were made with 500 μM NTP for single concentration measurements and 25–500 μM NTP for measurements establishing Michaelis-Menten kinetics. Reactions containing MurD were run with 500 μM GTP and MurD at the indicated concentration. Reactions were initiated with the addition of 5 μM protein and incubated at 37 °C for 90 min. Ten μL of reactions were quenched with 100 mM EDTA at 2 min, 30 min, 60 min, and 90 min, and inorganic phosphate was detected with the addition of 80 μL of 1 mM malachite green oxalate (Acros Organics), 10 mM ammonium molybdate (Sigma-Aldrich) in 1 M HCl. This sample mixture was incubated in the dark at ambient temperature for 10 min and then quenched with the addition of 10 μL 35% citric acid. Mixtures were incubated for 10 min and inorganic phosphate concentration was calculated based on the absorbance at 660 nm relative to a standard curve (5–500 μM phosphate, Fluka Analytical). Inorganic phosphate concentrations over time were used to calculate the rate for each reaction. Datapoints represent an average of 3 independently performed reactions.

### HPLC based assays of MurC and MurD activity

These reactions were run in 25 mM HEPES, 150 mM KCl, 2 mM $MgCl_2$, 2 mM TCEP, pH 7.4. UDP-MurNAc (UM) and UDP-MurNAc-*L*-Ala (UMA) were synthesized and purified from UDP-GlcNAc (Sigma Aldrich) following procedures modified from previously published protocols [71]. Michaelis-Menten initial velocity curves of MurD were determined by 200 μL reactions made with 200 μM UMA, 1 mM *D*-glutamic acid (Sigma Aldrich), and varying ATP from 25-500 μM. Reactions were initiated with 50 μL of a 4x enzyme mixture with a final reaction concentration of MurD to be 40 nM. In reactions containing Zn MigC, the 4x enzyme mixture contained 160 nM MurD, 16 μM Zn MigC (pre-loaded with 1.0 mol equivalent of $ZnSO_4$), and 500 μM GTP. For reactions containing bovine serum albumin (BSA), the 4x enzyme mixture contained 160 nM MurD, 0.61 mg/mL BSA (mg/mL equivalent to 16 μM Zn MigC), and 500 μM GTP. This enzyme mixture was incubated at 37 °C for 10 min prior to reaction initiation to allow sufficient GTP hydrolysis by Zn MigC. Reactions were incubated at 37 °C for 4 min and 45 uL was withdrawn and quenched with 5% formic acid at 1 min intervals. Precipitated protein was pelleted by centrifugation (10,000 rpm, 4 °C, 10 min) and sample remainder was filtered using a 0.22 μm filter Corning Spin-X nylon centrifuge tube filter (Millipore Sigma CLS8169). Ten μL of sample was injected on a Kinetex 5 μm XB-C18 100 Å, 150 x 4.6 mm (Phenomenex Inc., 00F-4605-E0) with a column guard (Phenomenex Inc., AJ0–8768) through an Agilent 1260 Infinity II HPLC. Analytes were separated using a gradient of buffer A (50 mM ammonium formate, pH 3.5) and buffer B (acetonitrile 0.1% formic acid): 0–5 min 0–10% Buffer B, 2 mL/min flow rate. MurD product, UDP-MurNAc-*L*-Ala-*D*-glu (UMAG), identified by monitoring absorbance at 254 nm and confirmed by ESI-MS (mass expected: 878.1751 *m/z*; mass observed: 878.1724 *m/z*) (S6 Fig). Concentration was quantified by peak area using the absorbance of the UDP group and a standard curve of UMA ranging from 0.5-500 μM. Rates of product formation were determined by calculating the slope of UMAG concentration over time. Parameters $V_{max}$ and $K_m$ were calculated by fitting data to the Michaelis-Menten equation using the Solver function in Excel [72].

Reactions designed to measure the relative activities of MurC and MurD were performed similarly. For MurC reactions, 200 μL reactions were made with 200 μM UM (UMA for MurD), 1 mM *L*-alanine (*D*-glutamate for MurD) (Sigma Aldrich), 1 mM ATP, and 0.1 mg/mL BSA. Reactions were initiated with 50 μL of a 4x enzyme mixture with a final reaction concentration of MurC or MurD to be 40 nM. For inhibition curves, the concentrations of Zn MigC were varied in the 4x enzyme mixture. 4x enzyme mixtures contained 160 nM MurC or MurD, 500 μM GTP, 0.1 mg/mL BSA, 0.1-300 μM apo or Zn MigC (pre-loaded with 1.0 mol equivalent of $ZnSO_4$). This enzyme mixture was incubated at 37 °C for 10 min prior to reaction initiation to allow sufficient GTP hydrolysis by Zn MigC. Reactions were performed following the same protocols of reactions determining MurD kinetics and analyzed by HPLC in the same way. UMA and UMAG peaks were fractionated and identified by ESI-MS (UMA mass expected: 749.1325 *m/z*; mass observed: 749.1306 *m/z*). Concentrations were determined by peak area as described above. Reaction activities were normalized to reactions containing the lowest concentration of

MigC. Inhibition curve calculations used final reaction concentrations of MigC ranging from 0.025-75 μM. Inhibition curves were simulated and fit to a four-parameter regression model (Equation 1) using the Solver function in Excel [73].

$$F(x) = V_i + \frac{V_u - V_i}{1 + \frac{x}{K_i}}$$

(1)

*Vi*: Inhibited activity of MurD in the presence of saturating MigC.
*Vu*: Uninhibited activity of MurD.
*Ki*: Dissociation constant for the enzyme-inhibitor complex.

The maximum inhibition of MurD ($V_i$) at saturating MigC was determined by a global fitting of experimental inhibition curves containing MigC and MigC variants. Using this maximal inhibition value as a fixed parameter, $K_i$ values were then determined separately for each condition.

## Computational tools

Sequence alignments and analysis were performed using CLC Genomics Workbench (version 24.0.1.) (Qiagen).

## Statistical analysis and quantification

All raw data were recorded initially in Microsoft Excel, imported to GraphPad Prism for statistical analysis, and then visualized using Envision Canvas software. Data were analyzed as indicated in Figure legends. Asterisks as determined through appropriate statistical analysis equivalate to: $*p < 0.05$, $**p < 0.01$, $***p < 0.001$, and $****p < 0.0001$. Data points of significant value are indicated in every Figure, with non-significant values not being represented. Specific statistical tests, significant values, dispersion, group sizes and precision of measurements are defined in the figure legends.

## Supporting information

**S1 Fig. AlphaFold3 Model of A1S_0934 predicts global structural changes upon nucleotide binding.** Ribbon diagram of AlphaFold3 model of A1S_0934 with one $Zn^{2+}$ ion bound, showing the G-domain (pale blue), C-terminal domain (grey), and the CxCC motif (inset) composed of C71, C73, and C74. Regions of conserved residues identified by SSN are denoted as G1 (cyan), G2 Switch I (orange), G3 Switch II (purple), G4 (royal blue), and G5 (yellow). The position of the highly conserved aspartic acid (D) to glutamic acid (E) substitution in the G3 loop (E99) of the protein is identified by the arrow.
(PDF)

**S2 Fig. A1S_0934 function may be affected by multiple divalent cations, which contributes to an elongated cell morphology.** Transmission electron microscopy was performed on WT or Δ*0934* cells in LB±40 μM TPEN. Cells were further assessed for (**A**) cell envelope, (**B**) inner, and (**C**) outer membrane width using ImageJ software. $*p < 0.05$ by one-way ANOVA.
(PDF)

**S3 Fig. MigC inhibits MurD activity under a variety of conditions.** (**A**) 1-by-1 interaction domain mapping yeast two-hybrid performed between full-length MigC and a truncated form of MurD, AA55–354, by Hybrigenics. Growth on agar with histidine supplemented indicates interactions between MigC and MurD. (**B**) Inhibition curves of MurD activity with varied apo MigC with (red) and without GTP (grey), Zn loaded MigC with GTP (blue), Zn loaded E99A MigC with GTP. A global curve fitting was performed to determine maximum inhibition of MurD at saturating MigC to be 75%. Curve fittings determined $K_i$ for apo MigC without GTP to be 20±3 μM, $K_i$ for apo MigC with GTP to be 43±7 μM, $K_i$ for Zn MigC with GTP to be 32±6 μM, and $K_i$ for Zn E99A MigC with GTP to be 21±6 μM.
(PDF)

**S4 Fig. Knocking down or overexpressing *murD* is detrimental for *A. baumannii* growth.** (**A**) WT *attTn7*::*dcas9*+*pyde007*::EV, and Δ*migC attTn7*::*Ptet*+pyde007::EV strains were grown in LB±50, 100, and 150 ng/mL of AhTc with $OD_{600}$ monitored over time. (**B**) WT *attTn7*::*Ptet-*+pYDE007-sgRNA$_{murD}$ were grown in LB±50, 100, and 150 ng/mL of AhTc with $OD_{600}$ monitored over time. (**C**) Δ*migC attTn7*::*Ptet dcas9*+pYDE007-sgRNA$_{murD}$ were grown in LB±50, 100, and 150 ng/mL of AhTc with $OD_{600}$ monitored over time. (**D**) WT *attTn7*::*Ptet-dcas9*+pYDE007 (EV), Δ*migC attTn7*::*Ptet-dcas9*+pYDE007 (EV), WT *attTn7*::*Ptet-dcas9*+pYDE007-sgRNA$_{murD}$, and Δ*migC attTn7*::*Ptet-dcas9*+pYDE007-sgRNA$_{murD}$±100 ng/mL AhTc were grown with $OD_{600}$ monitored over time. (**E**) These same strains were grown±50 ng/mL AhTc±20 µM TPEN with $OD_{600}$ monitored over time.
(PDF)

**S5 Fig. MigC contributes to *A. baumannii* CRO-susceptibility and virulence.** (**A**) WT and Δ*migC* integration controls and the complementation strain were grown for 20 hours in LB±7.5 µg/ml of CRO with $OD_{600}$ monitored over time. Data are represented as percent of LB growth at 8 hours. (**B-D**) Mice were intranasally infected with WT or Δ*migC* in mice given normal chow. Bacterial burdens were assessed at 36 hpi in the (**B**) kidney, (**C**) liver, or (**D**) spleen. Data are represented as the mean±SEM with each point indicating the bacterial burdens from an individual mouse in a specific organ. The limit of detection is indicated as LOD. \*\*\*$p < 0.001$ by unpaired *t* test.
(PDF)

**S6 Fig. MurD reaction specificity remains unchanged by MigC.** (**A**) HPLC chromatograms of representative reactions without MurD (black), with MurD (light grey), and with MurD and MigC (blue). (**B**) Chemical standards of GDP (black), ADP (dark grey), ATP (light grey), UMA (cyan), and UMAG (blue) used to identify peaks in MurD reaction chromatograms.
(PDF)

**S1 Table. Best-fit simulated parameters from the analysis of the Zn K-edge raw EXAFS for Zn$^{II}$-MigC.** [a]Data were fitted only over a k-range of 1–12.6 Å due to monochromator imperfections. [b]Independent metal-ligand scattering environment. [c]Scattering atoms: C (carbon), N (nitrogen) and O (oxygen). [d]Average metal-ligand bond length. [e]Average metal-ligand coordination number. [f]Average Debye-Waller factor in $Å^2$ x $10^3$. [g]Number of degrees of freedom weighted mean square deviation between data and fit.
(PDF)

**S2 Table. Strains used in this study [74–78].**
(PDF)

**S3 Table. Oligonucleotides used in this study.**
(PDF)

## Acknowledgments

We thank members of the Skaar and Giedroc laboratories for reviewing this manuscript.

## Author contributions

**Conceptualization:** Jeanette M. Critchlow, Joseph S. Rocchio, David P. Giedroc, Eric P. Skaar.

**Data curation:** Jeanette M. Critchlow, Joseph S. Rocchio, Melanie C. McKell, Courtney J. Campbell, Evan S. Krystofiak.

**Formal analysis:** Jeanette M. Critchlow, Joseph S. Rocchio, Melanie C. McKell, Courtney J. Campbell, Juan P. Barraza.

**Funding acquisition:** Jeanette M. Critchlow, Walter J. Chazin, David P. Giedroc, Eric P. Skaar.

**Investigation:** Jeanette M. Critchlow, Joseph S. Rocchio, Erin R. Green, Tae Akizuki, Walter J. Chazin, Michael S. VanNieuwenhze, Timmothy L. Stemmler, David P. Giedroc, Eric P. Skaar.

**Methodology:** Jeanette M. Critchlow, Joseph S. Rocchio, Erin R. Green, Eric P. Skaar.

**Project administration:** David P. Giedroc, Eric P. Skaar.

**Resources:** David P. Giedroc, Eric P. Skaar.

**Software:** Eric P. Skaar.

**Supervision:** David P. Giedroc, Eric P. Skaar.

**Validation:** Jeanette M. Critchlow, Joseph S. Rocchio.

**Visualization:** Jeanette M. Critchlow, Joseph S. Rocchio, Juan P. Barraza, Evan S. Krystofiak.

**Writing – original draft:** Jeanette M. Critchlow, Joseph S. Rocchio, Timmothy L. Stemmler, David P. Giedroc, Eric P. Skaar.

**Writing – review & editing:** Jeanette M. Critchlow, Joseph S. Rocchio, Courtney J. Campbell, Erin R. Green, Walter J. Chazin, Timmothy L. Stemmler, David P. Giedroc, Eric P. Skaar.

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
