## [Decision Letter · Decision Letter 0]

PPATHOGENS-D-24-02699

The Zinc Metalloprotein MigC Impacts Cell Wall Biogenesis through Interactions with an Essential Mur Ligase in Acinetobacter baumannii

PLOS Pathogens

Dear Dr. Skaar,

Thank you for submitting your manuscript to PLOS Pathogens. After careful consideration, we feel that it has merit but does not fully meet PLOS Pathogens's publication criteria as it currently stands. Therefore, we invite you to submit a revised version of the manuscript that addresses the points raised during the review process. Three reviewers and the editors have carefully considered this manuscript and all found the work in general to have many strengths and a large potential impact. However, the reviewers and the editor note that significant improvements are necessary to how the bacterial growth/fitness data are analyzed and presented. Reviewer 3 also recommends a more appropriate control for the murD overexpression experiment. Additional points to be addressed at minimum via textual changes are to consider an alternative model to explain MigC inhibition of MurD (see reviewer 3 comments), and integration of the findings with previous studies (see reviewer 2 comments).

Please submit your revised manuscript within 60 days Mar 15 2025 11:59PM. If you will need more time than this to complete your revisions, please reply to this message or contact the journal office at plospathogens@plos.org. Please include the following items when submitting your revised manuscript:

We look forward to receiving your revised manuscript.

Kind regards,

Edward Geisinger

Guest Editor

PLOS Pathogens

Matthew Wolfgang

Section Editor

PLOS Pathogens Sumita Bhaduri-McIntosh

Editor-in-Chief

PLOS Pathogens

orcid.org/0000-0003-2946-9497

Michael Malim

Editor-in-Chief

PLOS Pathogens

orcid.org/0000-0002-7699-2064

**Additional Editor Comments :**

In addition to the specific concerns raised by the reviewers regarding presentation of growth data, which I agree with, please also note that the last two labels appear swapped in Fig. 3D, and some labels are missing in Fig. 5H.

**Journal Requirements:**

At this stage, the following Authors/Authors require contributions: Jeanette Marie Critchlow. Please ensure that the full contributions of each author are acknowledged in the "Add/Edit/Remove Authors" section of our submission form.

- TM on page: 22 line 503.

5) We have noticed that you have uploaded Supporting Information files, but you have not included a list of legends. Please add a full list of legends for your Supporting Information files after the references list.

6) We note that your Data Availability Statement is currently as follows: "All relevant data are within the manuscript and its Supporting Information files. Additional data related to this study may be available from the corresponding author upon request.". Please confirm at this time whether or not your submission contains all raw data required to replicate the results of your study. Authors must share the “minimal data set” for their submission. PLOS defines the minimal data set to consist of the data required to replicate all study findings reported in the article, as well as related metadata and methods (https://journals.plos.org/plosone/s/data-availability#loc-minimal-data-set-definition).

7) Please amend your detailed Financial Disclosure statement. This is published with the article. It must therefore be completed in full sentences and contain the exact wording you wish to be published.

1) State what role the funders took in the study. If the funders had no role in your study, please state: "The funders had no role in study design, data collection and analysis, decision to publish, or preparation of the manuscript.".

8) Please ensure that the funders and grant numbers match between the Financial Disclosure field and the Funding Information tab in your submission form. Note that the funders must be provided in the same order in both places as well. Currently, the order of the funders is different in both places.. 

Please indicate by return email the full and correct funding information for your study and confirm the order in which funding contributions should appear. Please be sure to indicate whether the funders played any role in the study design, data collection and analysis, decision to publish, or preparation of the manuscript.

**Reviewers' Comments:**

Reviewer's Responses to Questions

**Part I - Summary**

Reviewer #1: The authors used an incisive series of carefully performed experiments to identify a non-canonical role for an Acinetobacter baumannii zinc-binding GTPase, initially predicted to be a metal chaperone, as an inhibitor of MurD. They confirmed that MigC hydrolyzes GTP (and it more weakly hydrolyzes ATP) in a manner that is stimulated by zinc binding. They used EXAFS spectroscopy to demonstrate tetrahedral coordination of zinc to three thiolate residues. Deletion of the gene encoding MigC leads to slower growth than for WT cells when treated with a chelator or in the presence of calprotectin, and these cells are elongated as shown by TEM. Yeast two-hybrid studies revealed clear evidence for MigC binding to MurD (with only minor effects on the affinity by zinc or GTP), and this interaction reduces the activity of the Mur ligase by ~75%. Disrupting the interaction between the proteins by transcriptional knock down studies leads to several physiological perturbations. Overall, this study provides keen insights into the non-chaperone role of MigC in A. baumannii cells. The science detailed here is multi-dimensional, first rate, and very clearly described. Nevertheless, a few issues should be addressed.

Reviewer #2: The manuscript by Critchlow et al reports on the discovery of a novel zinc metalloprotein (named here MigC) that modulates cell wall synthesis in response to zinc starvation through direct interactions with a Mur ligase enzyme. Specifically, during zinc-replete conditions, MigC downregulates MurD activity through direct protein-protein interactions. Upon zinc starvation (or deletion of MigC), this inhibition is relieved, resulting in higher cell wall synthesis (which can be detrimental). Overall, this is an impactful, well-done and well-written study. All conclusions are supported by rigorous data, and all the proper controls are in place. I have very few minor comments about the presentation, and about putting the data in a larger context.

Reviewer #3: In the manuscript by Critchlow et al. the authors analyze an uncharacterized Orf from Acinetobacter baumannii that shows homology to known GTP-binding Zn chaperones. In the absence of MigC, the bacteria show sensitivity to Zn depletion, which is consistent with the chaperone idea. However, the paper describes an interesting series of experiments that indicates this family of proteins may have much broader function than originally proposed, as the authors demonstrate that this protein, called MigC binds directly and convincingly to MurD, altering its function. MurD enzyme, which catalyzes covalent incorporation of D-Glutamate into cell wall stem peptides, showed reduced activity toward incorporation of D-Glu into UDP-MurNAc-L-Ala using an HPLC assay in the presence of MigC, Zn and GTP. This surprising result was supported by mutant studies, which show that depletion of the essential murD gene, shows increased fitness in presence of deletion of migC. Morphological changes in the presence of migC, murD, and double mutants are all consistent with the biochemical characterization of MigC interfering with MurD activity.

**Part II – Major Issues: Key Experiments Required for Acceptance**

Reviewer #1: The manuscript does not have any major issues in my opinion.

Reviewer #2: none noted

Reviewer #3: 1. This is quite an interesting piece of work, and the biochemistry is quite cool, but frankly the growth curves are displayed in a way that is totally noninformative. I believe what the authors are saying at a gut level, but I really can’t figure out what is going on when they display logarithmic growth on a linear scale. It makes everything look like a long lag, I really can’t tell whether there are different growth rates or yields are altered, etc. Displaying growth curves in this way is kind of the original sin of bacterial physiology and prevent proper evaluation of the data. This has to be redisplayed properly and then the authors have to study what it says and present the work accordingly, mindful of calculating growth rates (division times) when appropriate.

2. My second point is the MurD overexpressor data (Fig. 5G)is not particularly convincing (as far as I can tell based on linear growth curves) because the proper controls were not used, and the particular control used (Supp. Fig. 4F?) indicates that ATC addition causes a defect in fitness. I think the simplest solution is to delete the experiment, but if they want to keep the experiment, the control should be an active site mutation in MurD. Furthermore, the control is an “integration control.” I think this is lab jargon, because I have no idea what this means. I suppose it means no dCAS9 at the Tn7 site, but I am not sure. Finally, showing a critical control in Supplement and the experiment in the main text is not appropriate. The experiment only makes sense if you can compare to control, and it’s a little difficult to flip between figures to determine if the experiment makes sense.

3. Although this is speculation, it seems like invoking MigC as an inhibitor is certainly stating what the data are providing, but one might why an inhibitor itself has to be regulated by GTP levels in the cell and Zn ions. One might wonder whether its role is something other than being an inhibitor, but rather, it might be a specificity determinant that ensures side reactions or inappropriate substrates are not targeted by MurD. An example might be thinking about proofreading functions with DNA polymerases. Proofreading reduces polymerization rates, but it would be inappropriate to call proofreading an inhibitor. Another way to look at this is to recall the effects of Mn on Mg-containing enzymes. Mn can increase reaction rates, increase binding affinities, etc., but at the cost of lost of specificity. Mg might result in lower reaction rates and lower binding affinities, but with the reward of increasing substrate specificity. One wonders if there are other interfering reactions that MurD can catalyze that could cause cell wall disruption, and the MigC acts to prevent these side reactions.

**Part III – Minor Issues: Editorial and Data Presentation Modifications**

Reviewer #1: 1. Does the binding of MurD to zinc-MigC affect the latter’s GTPase activity?

2. The symbols are very difficult to distinguish in the growth curves shown in Figs. 3, 5, and S4. This problem could be overcome by using different colors for each sample in the separate panels.

3. Figure 3: Panel A includes the undefined “IC” which should be clarified in the legend. Panel D uses the delta migC nomenclature before the MurD interaction was described at which time the gene was renamed; at this point in the text the delta 0934 term should be retained.

4. Anhydrotetracycline is incorrectly shown as “anhydrous tetracycline” 4 times in the main text and 7 times in the supplemental section. The text could be simplified and the incorrect version eliminated by using the abbreviation (AhTc) that was defined in line 290. Unfortunately, this abbreviation was replaced with “Anhtet” in lines 321, 323, 507.

5. In Figure 5, panels A and B indicate varied concentrations, but the substance being varied is not clear without reading the legend. How long were the cells grown for the images in panel C, or is this not important? The y-axis of panel E (Percent LB) must be corrected since the concentration of LB was not varied.

6. Minor items: Add a reference for “Gecos CLI” in line 165? Add “AlphaFold3” reference in line 169? Provide references for the zinc-binding affinities in line 179? Thermodynamic constants (such as K) should be italicized throughout the manuscript. Line 203: Do the EXAFS data provide insight into whether the O/N ligand is an imidazole? Line 224: WT was not defined. The usage for WT and delta 0934 varies between being adjectives and nouns, but consistent terminology should be utilized. Line 242, use medium, not media. Line 263: is it residues 55-354 or 53-103? Line 329: use medium, not media. Line 338: fix “but to not the liver”. Line 340: why not use the CRO abbreviation? Line 415: change pUK18 to pUC18. Line 475: use medium, not media. Line 488: replace the lab jargon “log” with exponential. Line 508: replace “Overnights” with “Overnight cultures”. Line 523: “AA55-354” could be clarified as “amino acids” followed by the residue numbers. Also, is it 55 or 53? Line 524: How do bacterial “kinetic” growth assays differ from bacterial growth assays? Line 652-653: change “fit with, corrected” to “fit, with correction”. Line 686: define BSA. The term “AA1-200” in line 1026 was not immediately clear and should be described more fully. Line 1027 should include a reference for Gecos. Line 1065: use medium, not media. Line 1069: change order to TEM rather than ETM? Line 1081: define BSA. Line 1098: replace logarithmic with exponential.

7. Supplementary material minor items: Line 4: delete the period so the remainder is part of a sentence. Line 7-8: Please rephrase “conserved characteristic … substitution”. Line 11: A verb is needed in the line beginning “Electron” and the order is more commonly seen as TEM, not ETM. Lines 29, 30, 32, and 39: insert “were” before “grown”, and insert “were grown” in line 36. Line 44: what is meant by “kinetically”?

Reviewer #2: - Line 154. Just for narrative purposes, could you write a few lines about why A1_0934 in particular was prioritized for further study?

- Line 134, the logic is slightly off. Just because a protein is uncharacterized, does not mean that it can serve roles beyond those that are annotated. Perhaps rephrase to “…largely uncharacterized. We hypothesized that some of these proteins may indeed serve other roles beyond metallochaperone…”

- Line 239 – another slight narrative issue. The logical flow from Zn sensitivity of a mutant to assessing cell morphology is unclear. The paper that is cited here seems to indicate some connection between zinc starvation and cell wall synthesis; perhaps this could be mentioned as a rationale for looking at cell shapes? Their previous work on a zinc-dependent peptidoglycan hydrolase could also be mentioned here (if zinc starvation induces an endopeptidase, one may expect morphology changes).

- Line 252 – I would tread a little lightly here and not call it cell wall turnover (which suggests dynamic measurements, but what was done here is to measure HADA stain intensity). Also, the data show an increase in fluorescence, which indicates either reduced turnover, or increased synthesis. Perhaps it is safest to just describe the phenotype here and call it “increased HADA signal” instead of “peptidoglycan turnover”

- Discussion should mention their previous paper showing that a peptidoglycan hydrolase is upregulated during zinc starvation, which the present data are highly consistent with – perhaps it is necessary to relieve inhibition of MurD, thereby increasing flux into PG synthesis, in order to compensate for increased cleavage by ZrlA? The data in Fig. 3F are consistent with this as well – why is the cell diameter of a ∆0934 strain treated with TPEN increased? After all, MurD inhibition is already reduced due to the mutant, and perhaps what zinc starvation adds is PG hydrolase activity?

- Fig. 5E – the Y-axis labeling is a bit confusing. Percent LB sounds like you used diluted or concentrated LB. Should this be “Plating efficiency (carb/no carb)” or similar?

Reviewer #3: (No Response)

PLOS authors have the option to publish the peer review history of their article (what does this mean? ). If published, this will include your full peer review and any attached files.

**Do you want your identity to be public for this peer review?** For information about this choice, including consent withdrawal, please see our Privacy Policy .

Reviewer #1: No

Reviewer #2: No

Reviewer #3: No

**Figure resubmission:**
---

## [Editor Report · Decision Letter 1]

PPATHOGENS-D-24-02699R1

The Zinc Metalloprotein MigC Impacts Cell Wall Biogenesis through Interactions with an Essential Mur Ligase in Acinetobacter baumannii.

PLOS Pathogens

Dear Dr. Skaar,

Thank you for submitting your manuscript to PLOS Pathogens. After careful consideration, we feel that it has merit but does not fully meet PLOS Pathogens's publication criteria as it currently stands. Therefore, we invite you to submit a revised version of the manuscript that addresses the points raised during the review process.

Please submit your revised manuscript within 30 days May 27 2025 11:59PM. If you will need more time than this to complete your revisions, please reply to this message or contact the journal office at plospathogens@plos.org. Please include the following items when submitting your revised manuscript:

We look forward to receiving your revised manuscript.

Kind regards,

Edward Geisinger

Guest Editor

PLOS Pathogens

Matthew Wolfgang

Section Editor

PLOS Pathogens

Sumita Bhaduri-McIntosh

Editor-in-Chief

PLOS Pathogens

orcid.org/0000-0003-2946-9497

Michael Malim

Editor-in-Chief

PLOS Pathogens

orcid.org/0000-0002-7699-2064

**Additional Editor Comments:**

The revised manuscript has addressed the reviewer concerns and is now stronger overall compared to the initial submission.  I have only a few, very minor comments:

1) The new text added in p. 14 and 15 should be reread and edited to ensure accuracy and clarity. Specifically:

-lines 313-315: This sentence feels like a non-sequitur, since the migC-murD double lesion has not yet been compared to murD. Perhaps move this after the next sentence ending in line 317.

-lines 326-329: states upon TPEN treatment, but if referring to Fig. 5D the data do not include TPEN. Please clarify (perhaps this meant to refer to AhTc)?

2) Regarding Fig. S4:

-Please correct the graph titles in panels S4B and S4C. Legend states sgRNAmurD, but titles state pYDE007.

-Please update S4D labels to indicate the sgRNA used (they all state just pYDE007).

3) Additional items:

-lines 511, 517, 520, 544: replace anhydrous/Anhtet with the correct abbreviation

-lines 1105-1107: states strains were ± 75 ng/mL AhTc and plated on carbenicillin to assess viability, but panel 5E graph axis is labelled plating efficiency (Carb/No Carb). Please clarify what this label means, since it is unclear how carb/no carb was involved, and the role of ± ahTc in the experiment appears to be left out from readout.

-lines 1108, 1155: is 20mM correct for TPEN concentration? Figure S4 panel labels show 20 µM.

**Journal Requirements:**

We have noticed that you have uploaded Supporting Information files, but you have not included a complete list of legends. Please add a full list of legends for your Supporting Information files after the references list.

**Reviewers' Comments:**

**Figure resubmission:**
---

## [Editor Report · Decision Letter 2]

Dear Dr. Skaar,

We are pleased to inform you that your manuscript 'The Zinc Metalloprotein MigC Impacts Cell Wall Biogenesis through Interactions with an Essential Mur Ligase in Acinetobacter baumannii.' has been provisionally accepted for publication in PLOS Pathogens.

Before your manuscript can be formally accepted you will need to complete some formatting changes, which you will receive in a follow up email. Please also address the two minor editor comments. A member of our team will be in touch with a set of requests.

Best regards,

Edward Geisinger

Guest Editor

PLOS Pathogens

Matthew Wolfgang

Section Editor

PLOS Pathogens

Sumita Bhaduri-McIntosh

Editor-in-Chief

PLOS Pathogens

orcid.org/0000-0003-2946-9497

Michael Malim

Editor-in-Chief

PLOS Pathogens

orcid.org/0000-0002-7699-2064

We appreciate your attention to the reviewer and editor comments. Please note two remaining minor items:

(1) References to murD overexpression remain in two instances and these should be deleted/revised, since the overexpression experiments were removed in the resubmission.

-lines 386-387 "Additionally, overexpression of murD in WT cells decreases fitness" should be deleted, and the rest of the sentence revised or deleted.

-line 1098 (S4 Fig title), delete "or overexpressing"

(2) Ensure that the correct file for the revised version of Fig S4 is uploaded (it was revised in the combined pdf, but not in the individual file).
---

## [Editor Report · Acceptance letter]

Dear Dr. Skaar,

We are delighted to inform you that your manuscript, "The Zinc Metalloprotein MigC Impacts Cell Wall Biogenesis through Interactions with an Essential Mur Ligase in Acinetobacter baumannii.," has been formally accepted for publication in PLOS Pathogens.

Best regards,

Sumita Bhaduri-McIntosh

Editor-in-Chief

PLOS Pathogens

orcid.org/0000-0003-2946-9497

Michael Malim

Editor-in-Chief

PLOS Pathogens

orcid.org/0000-0002-7699-2064